# On the origin of universal cell shape variability in confluent epithelial monolayers

**Souvik Sadhukhan, Saroj Kumar Nandi***

Tata Institute of Fundamental Research, Hyderabad, India

**Abstract** Cell shape is fundamental in biology. The average cell shape can influence crucial biological functions, such as cell fate and division orientation. But cell-to-cell shape variability is often regarded as noise. In contrast, recent works reveal that shape variability in diverse epithelial monolayers follows a nearly universal distribution. However, the origin and implications of this universality remain unclear. Here, assuming contractility and adhesion are crucial for cell shape, characterized via aspect ratio ($r$), we develop a mean-field analytical theory for shape variability. We find that all the system-specific details combine into a single parameter α that governs the probability distribution function (PDF) of $r$; this leads to a universal relation between the standard deviation and the average of $r$. The PDF for the scaled $r$ is not strictly but nearly universal. In addition, we obtain the scaled area distribution, described by the parameter μ. Information of α and μ together can distinguish the effects of changing physical conditions, such as maturation, on different system properties. We have verified the theory via simulations of two distinct models of epithelial monolayers and with existing experiments on diverse systems. We demonstrate that in a confluent monolayer, average shape determines both the shape variability and dynamics. Our results imply that cell shape distribution is inevitable, where a single parameter describes both statics and dynamics and provides a framework to analyze and compare diverse epithelial systems. In contrast to existing theories, our work shows that the universal properties are consequences of a mathematical property and should be valid in general, even in the fluid regime.

**\*For correspondence:**
saroj@tifrh.res.in

**Competing interest:** The authors declare that no competing interests exist.

## Editor's evaluation

In this important study, the authors unveil the reason for nearly universal shape fluctuations that have been reported earlier by theoretically analysing the energy of a confluent epithelial tissue. The comparison of their analytic results with simulations and experimental data is compelling, only the justification of the cell area distribution is somewhat incomplete. The manuscript is relevant for all people with an interest in tissue structure and dynamics.

## Introduction

D'Arcy Thompson argued, in his book *On Growth and Form*, physical principles could explain tissue packing and cell shape (*Thompson, 1917*). Shape formation of tissues and organs during embryogenesis is a long-standing, fascinating problem of developmental biology. Since cells are the functional units of a tissue, shapes in the organs must originate at the cellular level (*Paluch and Heisenberg, 2009*; *Wyatt et al., 2015*; *Pérez-González et al., 2021*; *Hannezo et al., 2013*). Cell shapes are vital in both health and disease. As cancer progresses (*Sailem and Bakal, 2017*; *Park et al., 2016*), as asthma advances (*Park et al., 2016*; *Park et al., 2015*; *Veerati et al., 2020*; *Atia et al., 2018*), as wounds heal (*Nnetu et al., 2012*; *Poujade et al., 2007*), as an embryo develops (*Farhadifar et al., 2007*;

*Atia et al., 2018*), cells progressively change their shape. Besides, cell shape may influence crucial biological functions, such as cell growth or selective programmed cell death (apoptosis) (*Chen et al., 1997*), the orientation of the mitotic plane (*Wyatt et al., 2015*; *Bosveld et al., 2016*; *Xiong et al., 2014*), stem cell lineage (*McBeath et al., 2004*; *Wang et al., 2011*), terminal differentiation (*Watt et al., 1988*; *Roskelley et al., 1994*), and division-coupled interspersion in many mammalian epithelia (*McKinley et al., 2018*). Moreover, the nuclear positioning mechanism in neuroepithelia depends on cell shape variation (*Yanakieva et al., 2019*). Thompson regarded cell-to-cell shape variability as a biologically unimportant noise *Thompson, 1917*; however, it is now known that shape variability is not an exception but a fundamental property of a confluent cellular monolayer (*Graner and Riveline, 2017*). In a seminal work, (*Atia et al., 2018*) showed that cell shape variability, quantified by the aspect ratio ($r$), follows virtually the same distribution across different epithelial systems. But, the origin of this near-universal behavior, and whether it is precisely universal, remains unclear.

Previous works have shown the similarity in dynamics between cellular monolayers and glassy systems (*Angelini et al., 2011*; *Park et al., 2015*; *Malinverno et al., 2017*; *Garcia et al., 2015*). Additionally, crucial insights into the dynamics of cellular systems are obtained via simplified model systems, treating cells as polygons (*Park et al., 2015*; *Bi et al., 2015*; *Bi et al., 2016*; *Chiang and Marenduzzo, 2016*; *Sadhukhan and Nandi, 2021*; *Sussman et al., 2018*). One of these models, the vertex model (VM), shows a rigidity transition akin to the jamming transition (*Bi et al., 2015*; *Merkel et al., 2019*). However, this transition does not exist in other confluent models, such as the Voronoi model (*Sussman and Merkel, 2018*) or the cellular Potts model (CPM) (*Sadhukhan and Nandi, 2021*). But, all three models are similar from the perspective of glass transition (*Bi et al., 2016*; *Sussman et al., 2018*; *Chiang and Marenduzzo, 2016*; *Sadhukhan and Nandi, 2021*). A point of clarification on the terminology is quintessential here. The jamming (rigidity) transition is a zero-temperature, zero-activity phenomenon. It is a genuine phase transition characterized via an order parameter. The observed shape index, $q$, the ratio of perimeter to the square root of cell area, is an order parameter of the jamming transition (*Park et al., 2015*; *Bi et al., 2015*; *Merkel et al., 2019*). By contrast, the glass transition refers to the extreme dynamical slowing down when the relaxation time, $\tau$, becomes a certain value, usually taken as $10 - 100s$ in experiments. It is not associated with any phase transition, and no static order parameter exists (*Berthier and Biroli, 2011*). Extensive research in the last couple of decades shows that although jamming and glass transitions can coexist, they are distinct phenomena controlled by entirely different physics (*Mari et al., 2009*; *Biroli and Garrahan, 2013*; *Berthier and Witten, 2009*; *Ikeda et al., 2012*). However, these terms are often used imprecisely while describing the dynamics of biological systems. The 'jamming transition' is sometimes used to describe the changing system behavior from fluid-like fast to solid-like slow dynamics (*Berthier et al., 2019*; *Atia et al., 2021*); this is actually the glass transition. Their synonymous use may lead to erroneous conclusions. Since both transitions exist in this field, distinguishing them is crucial to avoid confusion (*Park et al., 2015*; *Chiang and Marenduzzo, 2016*; *Sussman et al., 2018*; *Sadhukhan and Nandi, 2021*).

*Atia et al., 2018* argued that the universal behavior of aspect ratio in an epithelial monolayer is related to the jamming transition. In a jammed system, one can carry out a voronoi tesselation centering each particle and obtain the tesselated volume, $x$, of the individual particles. $x$ is known to follow a $k$-Gamma distribution, $P(x, k) = [k^k/\Gamma(k)]x^{k-1}\exp[-kx]$, where $\Gamma$ is the Gamma function and the distribution is characterised by a single parameter $k$ (*Aste and Di Matteo, 2008*). From this association, (*Atia et al., 2018*) conjectured the viability of describing the distribution of $r$ via $P(x, k)$. But, the analytical derivation of the $k$-Gamma distribution in granular packing (*Aste and Di Matteo, 2008*) relies on the fact that $x$ is additive, whereas, as the authors of *Atia et al., 2018* rightly point out, $r$, is not. Thus, there "*exists no rigorous basis for the applicability of the k-Gamma distribution*" (*Atia et al., 2018*). Yet, (*Atia et al., 2018*) and several subsequent works *Lin et al., 2018*; *Li et al., 2021*; *Kim et al., 2020*; *Wenzel and Voigt, 2021* have shown that the probability distribution function (PDF) of scaled aspect ratio, $r_s$ (defined in 'Aspects of universality'), in diverse biological and model systems can be described by $P(r_s, k)$, and the value of $k$ is nearly the same, around 2.5, for these diverse systems. Furthermore, the standard deviation, $sd$, vs the mean aspect ratio, $\bar{r}$, follows a universal relation (*Atia et al., 2018*; *Kim et al., 2020*; *Ilina et al., 2020*). What is the origin of this universality? What determines the value of $k \sim 2.5$? How is the PDF of $r$ related to the microscopic properties of a system? Answers to questions like these are crucial for deeper insights into the cell shape variability

and unveiling the implications of the universality. However, it requires an analytical theory that is rare in this field due to the inherent complexity of the problem and the presence of many-body interactions.

The existing theoretical works on cellular shapes assume solid like property of the system: specifically, the property that deformation of a solid is elastic in nature, that is the system resumes its shape when the external force is relaxed. For example, cell shape in the context of the jamming transition within the VM has been studied in *Czajkowski et al., 2018*. In a recent work, Li et al demonstrated the presence of a gamma distribution of $r$ in a wide variety of systems (*Li et al., 2021*). The analytical framework of *Li et al., 2021* related this distribution to Boltzmann-like features and the elastic nature of the system. However, the origin of the universal properties and what dictates the value of $k \sim 2.5$ remained unclear. Furthermore, in the context of cellular systems, the solid-like nature is only applicable deep in the glassy regime or in a jammed system (*Czajkowski et al., 2018*). But, most biological systems are fluid-like due to activity. The universal behavior in diverse systems seems to suggest that a more general mechanism must exist. In this work, we take a different approach. We develop a mean-field analytical theory for cell shape variability without consideration of solid-like nature or rigidity, and thus, our results should be applicable even in the fluid regime. Crucially, our work reveals that the origin of the universal features is a mathematical property.

The main results of this work are as follows: (1) We find that the aspect ratio distribution is described by a single parameter, α, containing all the system-specific parameters. Having a single parameter within the theory implies that $\bar{r}$ determines the distribution. We demonstrate this in Figure 4f, illustrating the predictive power of the theory. This also implies a universal relation between $sd$ and $\bar{r}$ (*Figure 1e*, Figure 3f, Figure 4d). (2) The PDF of $r_s$ is not strictly, but nearly, universal; $k \sim 2.5$ is a direct consequence of a mathematical property. The k-Gamma distribution for $r_s$ is obtained as a rough approximation of our analytical expression. Crucially, this nearly universal distribution of $r_s$ exists for systems even in the fluid regime; we demonstrate this by comparing our analytical results with our simulations (Figure 3c) and existing experimental data (Figure 4c). (3) We also obtain the PDF for the scaled area, $a$, and show that it is not universal, in contrast to what has been proposed elsewhere (*Wilk et al., 2014*; *Figure 2* and Figure 4e). (4) We demonstrate that simultaneous measurements of the PDFs for $a$ and $r$ can reveal the effects of changing physical conditions, such as maturation, on the individual model parameters. We have verified our theory via simulations of two distinct models of a confluent epithelial monolayer: the discrete lattice-based CPM on square and hexagonal lattices and the continuous VM (see Appendix 2 for details). Moreover, comparisons with existing experimental data on a wide variety of epithelial systems show excellent agreements.

One remarkable aspect of our work is as follows. It is often hard to control a specific property in a biological system as a perturbation can significantly affect other proteins. However, α includes all such effects. Therefore, even in the absence of detailed knowledge of the individual changes, our theory allows the characterization of different effects in diverse systems by treating α as a control parameter. Such a characterization can illuminate the mechanistic notion if functions and shapes are related irrespective of the molecular details. We further demonstrate in our simulations that α can also be a parameter for the dynamics. Thus, the same parameter describes both statics and dynamics, governs the origin and aspects of universality, and provides a framework to analyze and compare diverse epithelial systems.

## Results

### Analytical theory for the shape variability

Simplified model systems, representing cells as polygons, have been remarkably successful in describing both the static and dynamic aspects of an epithelial monolayer (*Honda, 1978*; *Honda and Eguchi, 1980*; *Farhadifar et al., 2007*; *Bi et al., 2014*; *Fletcher et al., 2014*; *Bi et al., 2015*; *Park et al., 2015*). The energy function, $\mathcal{H}$, governing these models is

$$\mathcal{H} = \sum_{i=1}^{N} \left[ \lambda_A (A_i - A_0)^2 + \lambda_P (P_i - P_0)^2 \right], \tag{1}$$

where $N$ is the number of cells, the first term constrains area, $A_i$, to a target area, $A_0$, determined by cell height and cell volume, with strength $\lambda_A$. Cell heights in experiments remain almost constant in an epithelial monolayer (*Farhadifar et al., 2007*). The second term describes cortical contractility

and adhesion (*Figure 1a*; *Bi et al., 2015*; *Farhadifar et al., 2007*; *Prost et al., 2015*). It constrains the perimeter, $P_i$, to the target perimeter $P_0$ with strength $\lambda_P$. The energy function, *Equation 1*, can be numerically studied (*Albert and Schwarz, 2016*) via different confluent models, such as the VM (*Farhadifar et al., 2007*; *Bi et al., 2015*), the Voronoi model (*Bi et al., 2016*; *Honda, 1978*), or more microscopic models such as the CPM (*Graner and Glazier, 1992*; *Hirashima et al., 2017*; *Hogeweg, 2000*) and the phase-field model (*Nonomura, 2012*; *Palmieri et al., 2015*; *Wenzel and Voigt, 2021*). For concreteness, we mostly focus our discussions below within the CPM that has been demonstrated to be more appropriate for variable cellular shapes and sizes (*Bosveld et al., 2016*). However, our analytical results should generally apply for a confluent system independent of the microscopic details of the models, and we have verified them via numerical simulations in both the VM and the CPM on a square and hexagonal lattice. Furthermore, we have neglected cell division and apoptosis in our simulations for the results presented in the main text; their rates are usually low in epithelial monolayers (*Poujade et al., 2007*; *Park et al., 2015*). For example, they are of the order of $10^{-2}$ per hour and per day, respectively, for an MDCK monolayer (*Puliafito, 2017*). Nevertheless, we show in Appendix 4, that the general conclusions of the theory remain unchanged when their rates are not very large.

Our starting point is the energy function, *Equation 1*, describing a confluent system of cells. We assume that the probability of a specific cellular configuration is given by a Boltzmann weight at an effective temperature, $T$ (see 'Details of the analytical calculation' for details). Note that $T$ in an active system includes contributions from all possible activities and the equilibrium temperature. An exact interpretation of $T$ depends on the system, and several definitions of $T$ exist, for example, the ratio of correlation to response function (*Nandi and Gov, 2018*; *Petrelli et al., 2020*; *Nandi and Gov, 2017*), from Einstein relation (*Szamel, 2014*), etc. The confluent models (such as the VM or the CPM) for epithelial systems, have two main variants: depending on the presence or absence of activity in the form of self-propulsion. The second variant represents equilibrium systems; $T$ is treated at the same footing as an equilibrium temperature and provides good agreements with experiments (*Glazier and Graner, 1993*; *Hirashima et al., 2017*; *Sussman et al., 2018*; *Fletcher et al., 2014*). Thus, the Boltzmann distribution is justified, at least within our simulations (Appendix 1). Excellent agreements of our results with experiments and analyses of the experimental systems in terms of an effective temperature (*Atia et al., 2018*; *Kim et al., 2020*) also validate this description. (*Li et al., 2021*) also finds a Boltzmann distribution description at an effective temperature applies to a wide variety of systems. An exact analytical calculation for the distribution of $r$ is impractical; therefore, we have made several simplifying assumptions. They are either motivated by or justified in our simulations. Here, we briefly discuss the main aspects of the calculation and relegate the technical details to the Materials and Methods ('Details of analytical calculation') . A detailed comparison of the analytical theory with our simulations and justifications of the assumptions are shown in 'Comparison with simulations'.

One crucial aspect of these model systems (such as the VM, the CPM, or the Voronoi model) of epithelial monolayer is the constraint of confluency, which is area fraction is unity at all times; it enters the problem via the area term in *Equation 1*. This constraint is an intricate mathematical problem rendering a direct analytical calculation impractical. However, we can bypass this difficulty and gain valuable insights into the distribution of $r$. First, a thin actomyosin layer, known as cortex, mainly governs the cellular mechanical properties (*Prost et al., 2015*). Therefore, the perimeter term should be dominant in determining shape. Second, shape fluctuation in these models can occur only via changes in the boundary. Third, $r$ being non-dimensional can vary independently of the cell area. Thus, in the regime of our interest, when the cells are compact objects (in contrast to being fractal-like in other regimes *Sadhukhan and Nandi, 2021*, see Appendix 2), we expect the area term in *Equation 1* to be not crucial in determining $r$. We have tested this assumption in two different ways in our simulations. First, if the area term is not paramount, the distribution of $r$ should not depend on $\lambda_A$; as detailed later (*Figure 2b* and *Figure 2—figure supplement 1*), this is indeed true. Second, in the regime of our interest, the distribution of $r$ of a single cell (treating the rest of the system as medium *Graner and Glazier, 1992*; *Glazier and Graner, 1993*) is nearly the same as that of a confluent system (*Figure 2a*). Moreover, as discussed below, the energy from the area term varies only slightly as $\lambda_A$ changes. Therefore, we assume that the area constraint is satisfied and not crucial for the aspect ratio distribution.

Then the energy function, *Equation 1*, becomes a sum of energies coming from individual cells. Since the perimeter of a cell is independent of that of others, we can concentrate on a particular cell, $i$, with energy

$$\mathcal{H}_P = \lambda_P P_i^2 - 2\lambda_P P_0 P_i, \tag{2}$$

where the first term represents contractility, and the second, effective adhesion. We have ignored the constant part as it does not affect any system properties. We first develop a coarse-grained description of cell-perimeter designating it via a set of representative points, as described in 'Details of the analytical calculation'. To calculate the aspect ratio, $r$, we first need to obtain the two radii of gyrations, $s_1$ and $s_2$, around the two principal axes (see 'Radius of gyration' for the definitions). Then $r = s_1/s_2$, considering $s_1 > s_2$ without loss of generality. However, a direct calculation of $s_1$ and $s_2$ is intricate due to their anisotropic natures. A slightly simpler calculation is possible for $s$, the radius of gyration around the center of mass, and we have $s^2 = s_1^2 + s_2^2$ (*Davis and Denton, 2018*). Therefore, we first obtain the distribution of $s^2$, $P(s^2)$, and then using this, obtain $P(r)$. As detailed in 'Details of the analytical calculation', using $s^2 = A(r + 1/r)$, with $A$ being the cell area, we obtain

$$P(r) = \frac{1}{\mathcal{N}} \left(r + \frac{1}{r}\right)^{3/2} \left(1 - \frac{1}{r^2}\right) e^{-\alpha\left(r + \frac{1}{r}\right)}, \tag{3}$$

where the normalization constant $\mathcal{N}$ is determined via the constraint that total probability must be unity and $\alpha \propto \lambda_P(1 - KP_0)/T$ with $K$ being a constant.

Additionally, as detailed in 'Distribution for area', *Equation 18* together with the constraint of confluency (*Weaire, 1986*; *Gezer et al., 2021*), give the distribution for the scaled area $a = A/\bar{A}$, where $\bar{A}$ is the average of area. It is a Gamma distribution, with a single parameter μ,

$$P(a) = \frac{\mu^\mu}{\Gamma(\mu)} a^{\mu-1} \exp[-\mu a]. \tag{4}$$

Since μ is related to the constraint of confluency, it should be independent of $\lambda_P$; our simulations show that this is indeed true (*Figure 3e*). Therefore, α and μ together can distinguish how the model parameters $\lambda_P$ and $T$ are affected by changing conditions such as maturation.

Note that the power 3/2 of the algebraic term in *Equation 3* comes from the mathematical property of closed-looped objects. That is, for closed-loop objects, the lowest non-zero eigenvalue will have degeneracy 2, leading to the exponent 3 in the algebraic part of *Equation 18*, as detailed in 'Details of the analytical calculation'. As shown below (next section), *Equation 3* can be roughly approximated as a $k$-Gamma distribution for $r_s$ that has been fitted with experimental and simulation data (*Atia et al., 2018*; *Kim et al., 2020*; *Li et al., 2021*; *Wenzel and Voigt, 2021*). The value of $k \sim 2.5$ found in these fits comes from this mathematical property. Since the perimeter of a cell must be closed-looped, this mathematical property is inevitable. On the other hand, all the system specific details are contained in the parameter $\alpha$. We treat it as a free parameter in the theory and obtain its value via fits with data. Thus, *Equation 3* provides a remarkable description, where all the system-specific details enter through a single parameter, α; it has profound implications leading to the universal behavior as we now illustrate.

## Aspects of universality

We show the PDF of the aspect ratio, $P(r)$, at different values of α in *Figure 1b*, $P(r)$ decays faster as $\alpha$ increases, as expected from *Equation 3*. The plots look remarkably similar to the experimental results shown in *Atia et al., 2018*; we present detailed comparisons with experiments later. (*Atia et al., 2018*) has demonstrated that the PDFs of the scaled aspect ratio, $r_s = (r - 1)/(\bar{r} - 1)$, where $\bar{r}$ is the ensemble-averaged $r$, across different systems follow a near-universal behavior. We now plot the PDFs of $r_s$ in *Figure 1c*. The PDFs *almost* overlap, but they are not identical. A closer look at *Equation 3* shows that if $\bar{r} + 1/\bar{r}$ goes as $1/\alpha$, we can scale $\alpha$ out of the equation and obtain a universal scaled distribution for $r_s$. However, as shown in *Figure 1d*, there is a slight deviation in $\bar{r} + 1/\bar{r}$ with the functional form of $1/\alpha$. This tiny deviation implies that the PDF of $r_s$ is not universal. If one ignores $1/r$ compared to $r$, *Equation 3* becomes a $k$-Gamma function for $r_s$ with $k = 2.5$. However, since $r \sim \mathcal{O}(1)$, this cannot be a good approximation, and the observed spread of $k$ around 2.5 is natural when fitted with this

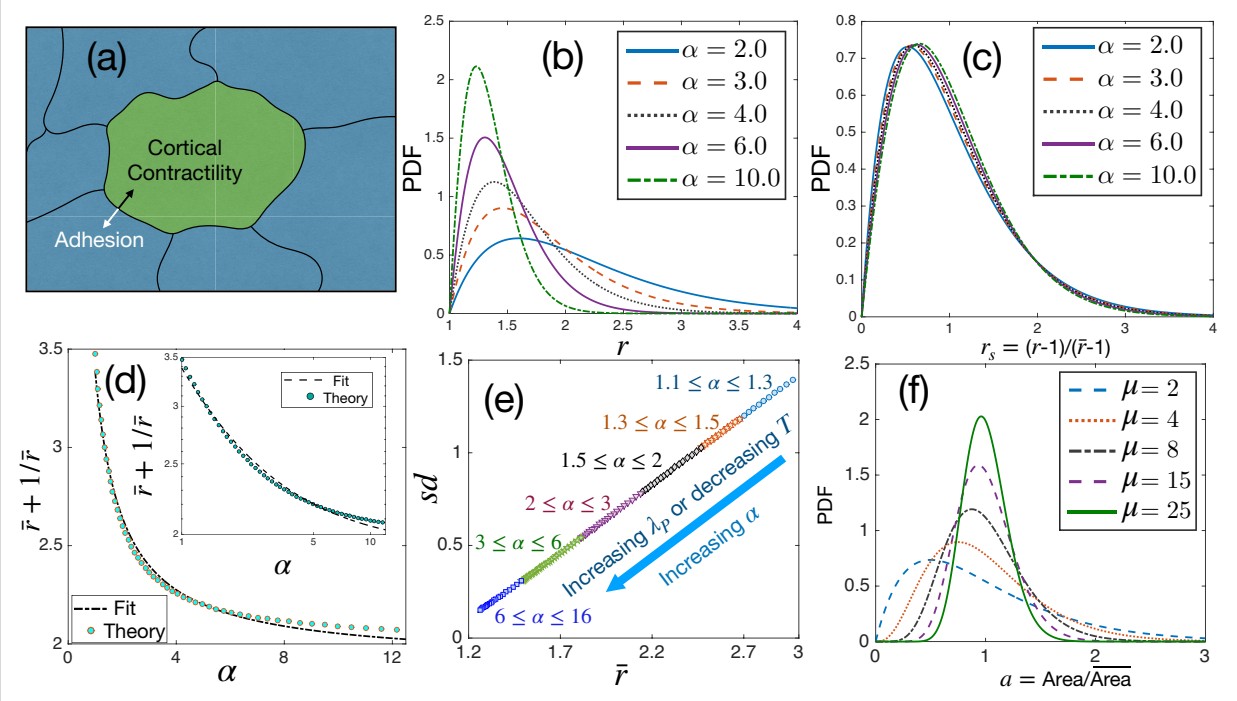

**Figure 1.** Theoretical results for cell shape variability. (**a**) Schematic illustration of a confluent model of an epithelial monolayer. Cortical contractility and adhesion impose competing forces. (**b**) PDF of aspect ratio, $r$, is governed by the parameter $\alpha$. (**c**) PDF of the scaled aspect ratio, $r_s = (r-1)/(\bar{r}-1)$ is nearly universal. (**d**) The PDF of $r_s$ can be universal if $\bar{r} + 1/\bar{r}$ is proportional to $1/\alpha$ (**Equation 3**), but there is a slight deviation showing that it is not strictly universal. Inset: Same as in the main figure, but in log-log scale. Lines are fits with the $1/\alpha$ form. (**e**) Standard deviation ($sd$) vs $\bar{r}$ follows a universal relation; the state points move towards the origin as $\alpha$ increases. (**f**) The PDF of $a$ follows Gamma distribution with a single parameter, $\mu$, **Equation 4**.

function (**Atia et al., 2018**; **Li and Ciamarra, 2018**; **Kim et al., 2020**; **Wenzel and Voigt, 2021**). On the other hand, since the deviation (**Figure 1d**) is minute, the PDFs of $r_s$ for different systems look nearly universal (**Figure 1c**): the $\alpha$-dependence becomes so weak for the PDFs of $r_s$ that they seem to be independent of $\alpha$. This result is a strong prediction of the theory and, as we show below, is corroborated by available experimental data on diverse epithelial systems.

Although the PDFs of $r_s$ are not strictly universal, there is another aspect, $sd$ vs $\bar{r}$, which is universal. We show the $sd = \sqrt{\overline{r^2} - \bar{r}^2}$ as a function of $\bar{r}$ in **Figure 1e**. Since there is only one parameter, $\alpha$, in **Equation 3**, it determines both $sd$ and $\bar{r}$. The monotonic dependence of $\bar{r}$ on $\alpha$ (**Figure 1d**) implies a unique relationship between them. Therefore, we can express $\alpha$ in terms of $\bar{r}$ and, in turn, $sd$ as a function of $\bar{r}$. Since there is no other system-dependent parameter in this relation, it must be universal. Note that $\alpha \propto \lambda_P/T$ at a constant $P_0$, thus, $\alpha$ increases as $\lambda_P$ increases or $T$ decreases. Both $sd$ and $\bar{r}$ become smaller as $\alpha$ increases, and the system on the $sd$ vs $\bar{r}$ plot moves towards the origin (**Figure 1e**). From the perspective of the dynamics, the relaxation time, $\tau$, of the system grows as $\alpha$ increases (**Sadhukhan and Nandi, 2021**). Thus, small $\bar{r}$ and large $\tau$, that is, less elongated cells and slow dynamics, follow each other, and the energy function, **Equation 1**, controls both behaviors. Finally, we show in **Figure 1f** some representative PDFs, **Equation 4**, at different values of $\mu$ for the scaled area $a = A/\bar{A}$. The PDF of $a$ has been argued to be universal **Wilk et al., 2014**; our theory shows that although the PDFs of $a$ for different systems follow the same functional form, they are not identical.

## Comparison with simulations

We now compare our analytical theory with simulations of two distinct confluent models: the CPM and the VM. In our simulations, we use the original energy function, **Equation 1**, and other simulation details are presented in the Appendix 2. Unless otherwise stated, the CPM simulations, presented in the main text, are on the square lattice; the data for hexagonal lattice CPM simulations are shown in **Figure 3—figure supplement 1**. We first present tests of the crucial assumption that, in the regime of

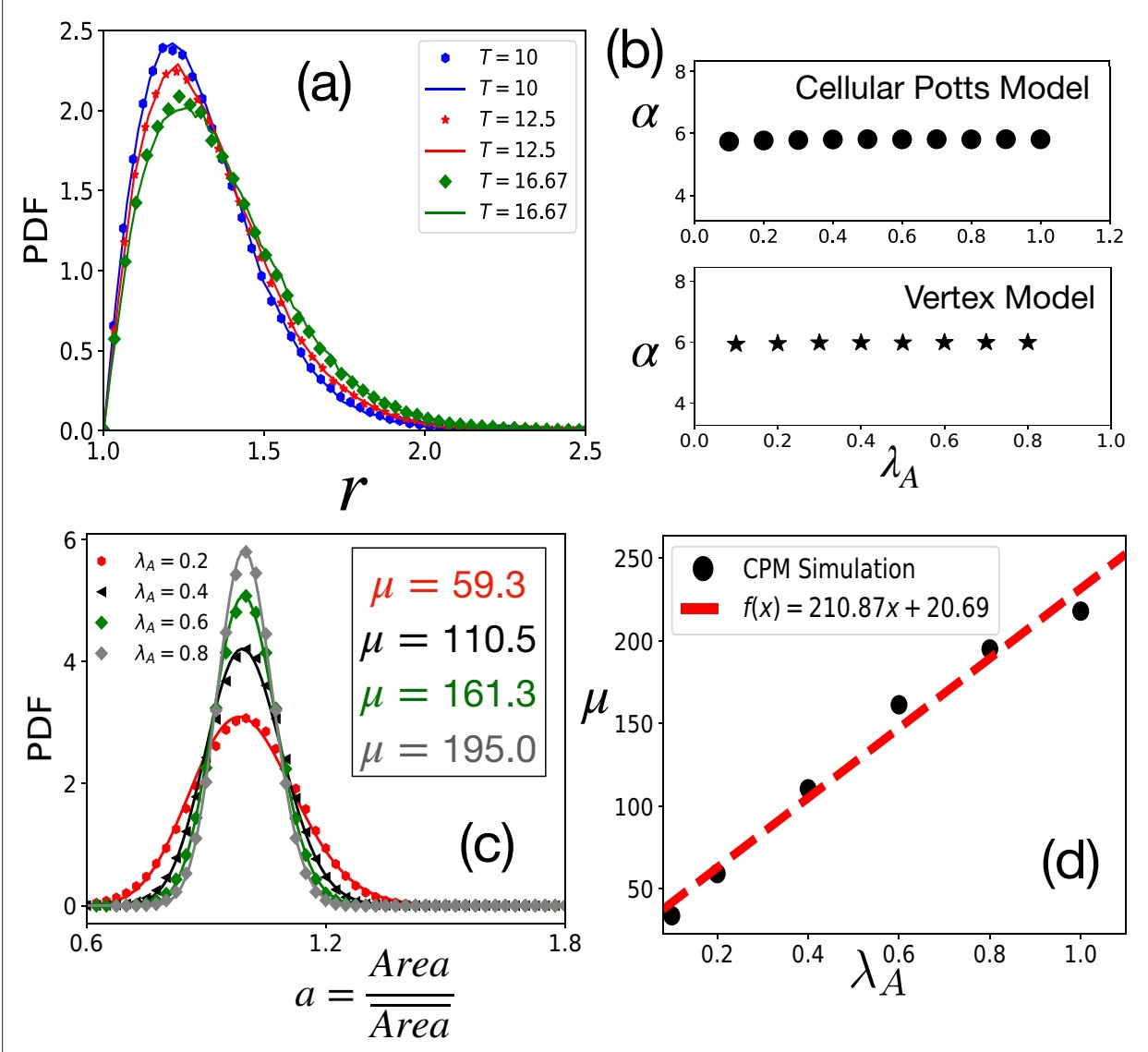

**Figure 2.** Tests of theoretical assumptions. (**a**) PDFs of aspect ratio, $r$, within the CPM for a single cell in a medium are nearly identical with those for a confluent system when the parameters are the same: $\lambda_A = 1.0$, $\lambda_P = 0.5$, $P_0 = 38$ and $A_0 = 90$. Symbols and lines are data for the confluent system and single cell in a medium, respectively. (**b**) $\alpha$, giving the distribution of $r$ remains independent of $\lambda_A$. For the CPM, $\lambda_P = 0.5$, and $T = 25.0$; for the VM $T = 2.5 \times 10^{-3}$, $\lambda_P = 0.02$, $A_0 = 1$, and $P_0 = 3.7$. (**c**) PDF of the scaled area, $a$, within the CPM at different $\lambda_A$, but fixed $T = 12.5$ and $\lambda_P = 0.5$. The lines are fits with *Equation 4* with the values of μ as shown. (**d**) μ almost linearly increases with $\lambda_A$. For the CPM simulations in (**b**) and (**c**), we have $A_0 = 40$ and $P_0 = 26$. The PDFs are calculated from at least $4 \times 10^4$ independent configurations.

The online version of this article includes the following figure supplement(s) for figure 2:

**Figure supplement 1.** Distribution of aspect ratio does not depend on $\lambda_A$.

**Figure supplement 2.** Distribution of aspect ratio for different values of $\lambda_A$.

**Figure supplement 3.** Scaled area distribution within the VM with varying $\lambda_A$.

**Figure supplement 4.** Dependence of μ on $T$.

our interest here, we can ignore the constraint of confluency that enters via the area term in *Equation 1*. We simulate, within the CPM, single cells treating the rest of the system as medium and compare the distribution of $r$ with that in a confluent system. As shown in *Figure 2a*, the PDFs are nearly the same. Next, we have simulated the confluent systems with varying $\lambda_A$ and find that the distribution of $r$ remains almost independent of $\lambda_A$. We have obtained $\alpha$ that characterizes the aspect ratio distribution via *Equation 3* at fixed $\lambda_P$, $P_0$, and $T$ but varying $\lambda_A$. As shown in *Figure 2b*, within both the CPM and

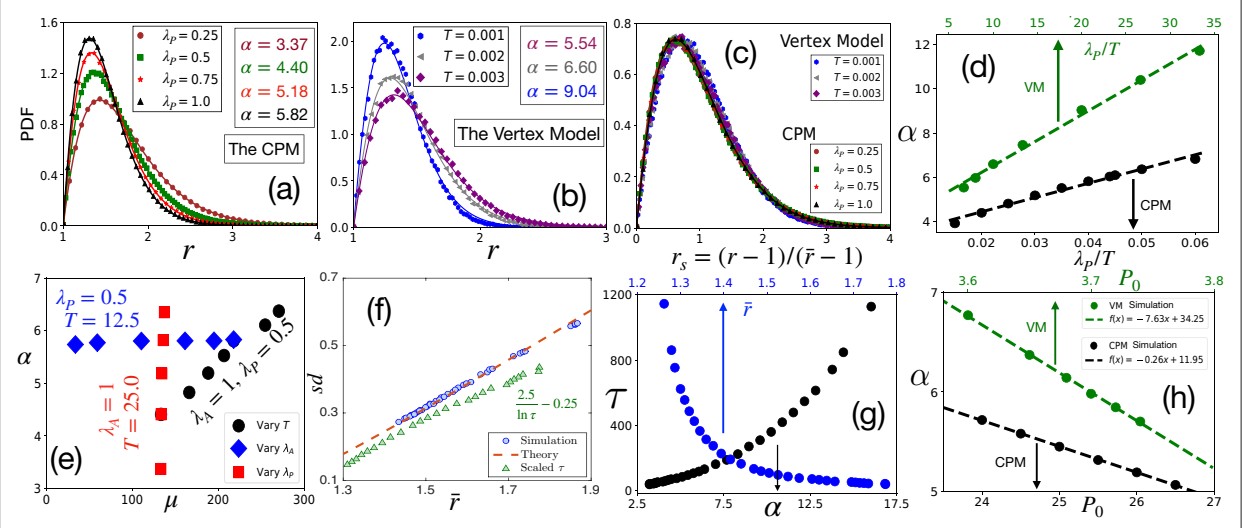

**Figure 3.** Comparison with simulations. (**a**) PDF of $r$ at different $\lambda_P$ within the CPM, with $\lambda_A = 1.0$, and $T = 25.0$. Symbols are simulation data, and lines represent the fits with *Equation 3*. (**b**) PDF of $r$ within the VM with varying $T$, $\lambda_P = 0.02$, $\lambda_A = 0.5$, $P_0 = 3.7$. Lines are fits with *Equation 3* with $\alpha$ as shown. (**c**) PDFs for the $r_s$, for the same data as in (**a**) and (**b**) for the CPM and the VM, respectively, show a virtually universal behavior. (**d**) Theory predicts $\alpha$ linearly varies with $\lambda_P/T$, simulation data within both the models agree with this prediction. $\lambda_A = 1.0$ for the CPM; $P_0 = 3.7$ and $\lambda_A = 0.5$ for the VM simulations. Dotted lines are fits with a linear form. (**e**) $\alpha$ vs $\mu$ within the CPM when we have varied only one of the three variables: $T$, $\lambda_A$, and $\lambda_P$ does not depend on $\lambda_A$, and $\mu$ does not depend on $\lambda_P$. (**f**) Our theory predicts a universal behavior for $sd$ vs $\bar{r}$, symbols are the CPM data at different parameter values, and the line is our theory (not a fit). We have also plotted the scaled relaxation time, $2.5/\ln \tau - 0.25$, to show them on the same figure. $\tau$ increases as $\bar{r}$ decreases. (**g**) $\tau$ as functions of $\alpha$ (lower axis) and $\bar{r}$ (upper axis). (**h**) Theory predicts $\alpha$ linearly decreases with $P_0$, this agrees with simulations (symbols); parameters in the CPM: $\lambda_A = 1.0$, $\lambda_P = 0.5$, $T = 16.67$; the VM: $\lambda_A = 0.5$, $\lambda_P = 0.02$, $T = 0.0025$. [CPM simulations here are on square lattice, $A_0 = 40$, $P_0 = 26$ in (**a–f**); PDFs are calculated over at least $4 \times 10^4$ independent configurations].

The online version of this article includes the following figure supplement(s) for figure 3:

**Figure supplement 1.** Distribution of aspect ratio is independent of lattice type.

**Figure supplement 2.** Dependence of $\alpha$ on $\lambda_P$ and $T$ within the CPM and the VM.

the VM, $\alpha$ remains almost constant with varying $\lambda_A$ (the distributions are shown in *Figure 2—figure supplement 2*). Finally, the scaled area, $a$, given by *Equation 4*, is a sharply peaked function around $a = 1$. We show the distribution of $a$ within the CPM in *Figure 2c*: the lines are fits with *Equation 4* with $\mu$ as a fitting parameter. Thus, assuming that cells satisfy the target area, we can ignore the area term in *Equation 1* to obtain the distribution of $r$.

*Figure 2c* shows that $P(a)$ becomes more sharply peaked as $\lambda_A$ increases; this makes sense as greater values of $\lambda_A$ ensure the area constraint is more effective and the distributions become sharply peaked around the average area. Thus, the standard deviation of $A$ decreases as $\lambda_A$ increases. This implies that the variation in energy from the area term in *Equation 1* is even less. We have checked that it is less than 10% for about a 300% change in $\lambda_A$. Thus, treating the area part of *Equation 1* as a constant is justified. *Figure 2d* shows that $\mu$ almost linearly increases with $\lambda_A$. We present similar results for the VM in *Figure 2—figure supplement 3*. Since the area term is related to the cell height that remains nearly constant and the geometric constraint of confluency, we do not expect a substantial variation in $\lambda_A$ in a particular system. However, $\mu$ also varies with $T$ (see *Figure 2—figure supplement 4*). Thus, in contrast to what has been proposed elsewhere (*Wilk et al., 2014*), as $T$ changes, though the PDF of $a$ is well-described by the same function, *Equation 4*, the values of $\mu$ can be different. Thus, the PDFs for different systems or the same system at different levels of activity and maturation need not be identical.

We now show that our analytical theory agrees well with simulation data. To highlight that the distribution of $r$ and its associated universal properties are also valid in the fluid regime, we have mostly simulated the systems in this regime. However, some of the simulations are also in the glassy regime (with relaxation time greater than $10^4$). Since glass transition is not associated with any thermo-dynamic transition, we do not expect a drastic change in the static properties, such as the distribution of $r$. *Figure 3a and b* show representative plots for the comparison of the PDFs of $r$ within the CPM,

and the VM, respectively, where the lines represent fits with *Equation 3*. *Figure 3a* shows data with varying $\lambda_P$, and *Figure 3b* shows data with changing $T$. As discussed above, our theory predicts nearly universal behavior for the PDFs of $r_s$ (*Figure 1c*). We plot the simulation data for the PDFs of $r_s$ for both the models at different parameters in *Figure 3c*; the PDFs almost overlap, consistent with the theory (see *Figure 3—figure supplement 1* for more results). An important prediction of the theory is that the parameter $\alpha$, which governs the behavior of the cell shape variability, is linearly proportional to both $\lambda_P$ and $1/T$, hence with $\lambda_P/T$. This prediction also agrees with our simulations (*Figure 3d* and *Figure 3—figure supplement 2*). The slopes within the CPM and the VM are different; this possibly comes from the distinctive natures of the two models, but the qualitative behaviors are the same.

*Figure 3e* shows α vs μ within the CPM when we vary one of the parameters, $\lambda_A$, $\lambda_P$, and $T$, keeping the other two fixed. First, when $\lambda_A$ increases, the value of μ increases, but α remains almost constant (also see *Figure 2b*). Next, when $\lambda_P$ increases, although α linearly increases, μ remains nearly the same. Finally, both parameters linearly increase with $1/T$; since higher $T$ implies more fluctuations, decreasing $T$ helps both $r$ and $a$ to become sharply peaked (see *Figure 2—figure supplement 4*, *Figure 3—figure supplement 2* for their specific behaviors, and results within the VM). These results show when $\lambda_A$ remains constant, varying $\lambda_P$ and $T$ have distinctive effects on μ and α. These results are significant from at least two aspects: First, μ comes from the constraint of confluency (see 'Distribution for area'), which should depend only on the area and be independent of the perimeter. Thus, the $\lambda_P$-independence of μ validates the phenomenological implementation *Weaire, 1986*; *Gezer et al., 2021* of this constraint. Second, these results can provide crucial insights regarding the model parameters. The maturation of a monolayer can affect both $\lambda_P$ and $T$. Additional junctional proteins may be employed during maturation to increase $\lambda_P$. On the other hand, different forms of activity may reduce, decreasing $T$. Since $\alpha$ increases linearly with $\lambda_P/T$, $r$ alone is not enough to determine the dominant mechanism during the maturation process. However, assuming that $\lambda_A$ remains constant in a particular system, simultaneous measurements of μ and α allow distinguishing effects of changing physical conditions, such as maturation, on the individual parameters.

We next verify the universal result of the theory: $sd$ vs $\bar{r}$. *Figure 3f* shows $sd$ vs $\bar{r}$ within the CPM; we plot the theoretical prediction by the dotted line for comparison. The theory predicts that the state points move towards the origin as $\alpha$ increases (*Figure 1e*); this is consistent with our simulations. Since $\alpha \propto \lambda_P/T$, higher $\alpha$ should correspond to slower dynamics. To test this hypothesis, we have simulated the CPM at different $\lambda_P$, $P_0$, and $T$ to obtain the relaxation time, $\tau$ (see Appendix 2 for details). From these control parameters, we have calculated $\alpha$ and then $\bar{r}$, using our theory. We show $\tau$ as functions of $\alpha$ and $\bar{r}$ in *Figure 3g*; it is clear that indeed $\tau$ grows as $\alpha$ increases or $\bar{r}$ decreases. To show this behavior of $\tau$ on the same plot as $sd$ vs $\bar{r}$, we plot $2.5/\ln \tau - 0.25$ in *Figure 3f* as a function of $\bar{r}$. A monolayer fluidizes under compressive or stretching experiments, where cell shape changes, but not cell area (*Krishnan et al., 2012*; *Park et al., 2015*; *Atia et al., 2018*). Such perturbations make the cells more elongated, increasing $\bar{r}$; thus, our theory rationalizes the decrease in $\tau$ associated with fluidization under such perturbations. Finally, we show that our mean-field result that $\alpha$ decreases linearly with $P_0$ agrees with simulations (*Figure 3h*). Further, to test our hypothesis that our main results remain unchanged in the presence of cell division and apoptosis, when their rates are small, we have simulated the CPM, including these processes. As shown in the Appendix 4, the simulation results justify our hypothesis.

## Comparison with existing experiments

Having shown that our theory agrees well with both the CPM and the VM simulations, we next confront it with the existing experimental data. We first compare the theory with data taken from *Atia et al., 2018* for three different confluent cell monolayers: the MDCK cells, the asthmatic HBEC, and the non-asthmatic HBEC. We chose the PDFs at three different times from Figure 3a and c in *Atia et al., 2018*. We fit *Equation 3* with the data to obtain $\alpha$ and present their values in *Table 1*; the corresponding fits for the MDCK cells are shown in *Figure 4a* (see *Figure 4—figure supplement 1* for the other fits). *Table 1* shows that α increases with maturation. Thus, progressive maturation can be interpreted as an increase in either $\lambda_P$ or $1/T$ or both. The PDFs for $r_s$ corresponding to the MDCK cells are shown in the inset of *Figure 4a* together with one set of experimental data *Atia et al., 2018*; note that this is not a fit, yet the theory agrees remarkably well with the data.

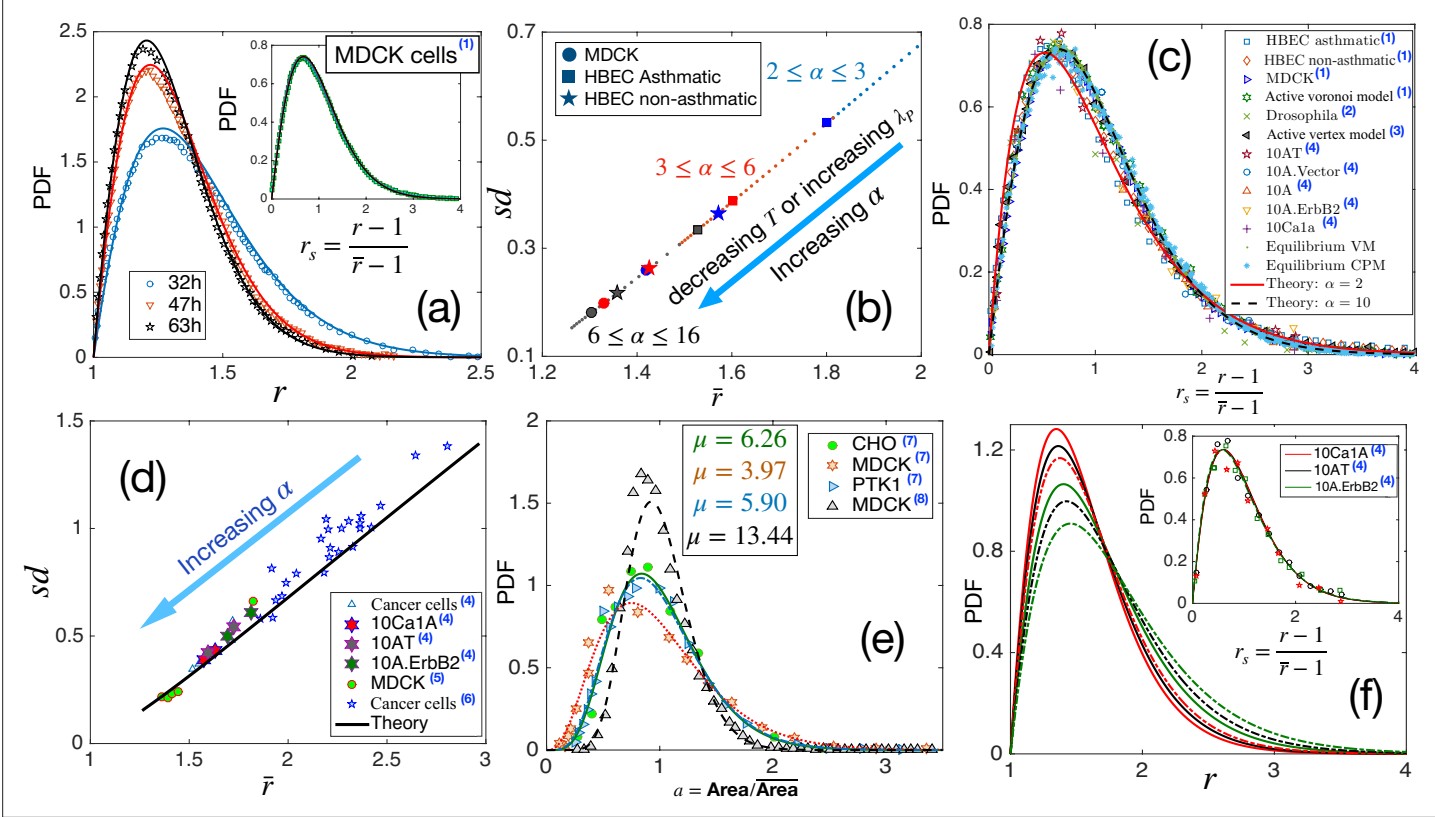

**Figure 4.** Comparison with existing experiments. (**a**) PDF for aspect ratio, $r$. Symbols are experimental data at three different times for MDCK cells, and lines represent the fits with our theory, *Equation 3*. The values of $\alpha$ are quoted in *Table 1*. Inset: PDF of the $r_s$, the lines are theory, and the symbols are data. (**b**) Using the values of $\alpha$, obtained for the three sets of data as quoted in *Table 1*, we obtain $sd$ vs $\bar{r}$ using our theory. Different symbols represent the types of the systems, and the colors blue, red, and black represent early, intermediate, and later time data, respectively. With maturation, the system moves towards lower $\bar{r}$ and smaller $sd$. (**c**) PDF for $r_s$ for a wide variety of systems, shown in the figure, seems to be nearly universal, consistent with our theory. (**d**) The theory predicts a universal relation for $sd$ vs $\bar{r}$. Symbols are data for different systems, and the line is our theoretical prediction. (**e**) PDF of the scaled area for different epithelial systems and the lines represent the fits with *Equation 4*. (**f**) Predictive power of the theory: we use the $\bar{r}$ for the three sets of cells from *Kim et al., 2020*, as marked by the hexagrams in (**d**), and obtain the corresponding values of $\alpha$ and obtain the PDFs for $r$ using *Equation 3*. The colors correspond to the type of cells in (**d**), and the continuous line corresponds to the lower $\bar{r}$ data. Inset: Lines are the theoretical PDFs for $r_s$ using the values of $\alpha$ for different cells, and the symbols show the experimental data. We have collected the experimental and some of the simulation data from different papers. Data taken from other papers are marked with a blue superscript in the legends. The sources are as follows: (1) *Atia et al., 2018*, (2) *Li et al., 2021*, (3) *Lin et al., 2018*, (4) *Kim et al., 2020*, (5) *Fujii et al., 2019*, (6) *Ilina et al., 2020*, (7) *Wilk et al., 2014* and (8) *Puliafito, 2017*.

The online version of this article includes the following figure supplement(s) for figure 4:

**Figure supplement 1.** Comparison of theory with experiments on HBEC cell monolayer data of *Atia et al., 2018*.

We next calculate $sd$ as a function of $\bar{r}$ using the values of $\alpha$, noted in *Table 1* for the three systems. They are shown in *Figure 4b* along with the theory prediction. With maturation, the state points move towards lower $\bar{r}$, represented by the arrow in *Figure 4b*. As shown in *Figure 3g*, larger $\alpha$ corresponds to a system with higher $\tau$. Thus, with maturation, as the PDFs become sharply peaked, as the cells look more roundish and $\bar{r}$ becomes smaller, the system becomes more sluggish. This maturation effect is the same in all the systems (*Figure 4b*) and agrees with the interpretation presented in *Atia et al., 2018*. We have also examined that the theoretical prediction of $sd$ vs $\bar{r}$ agrees well with the experimentally measured values shown in *Atia et al., 2018*.

Our theory predicts that the PDF for $r_s$, although not strictly universal, should be almost the same for different systems (*Figure 1c*). This prediction is a consequence of a crucial aspect of the theory: all the system-specific details enter via a single parameter, $\alpha$ in *Equation 3*. As shown in *Figure 1d*, $\bar{r} + 1/\bar{r}$ deviates slightly from the behavior $1/\alpha$. This slight deviation implies that the PDF for $r_s$ can not be strictly universal and manifests as a variation in k when the PDF is fitted with the k-Gamma

**Table 1.** Values of α from fits of *Equation 3* with the PDFs of $r$.
The data for the three systems are taken as function of maturation time, in units of h (hours) and d (days). The experimental data are taken from *Atia et al., 2018*.

| Cell type | Time | Value of $\alpha$ |
|---|---|---|
| | 32 h | 7.535 |
| MDCK | 47 h | 10.998 |
| | 63 h | 12.496 |
| | 6 d | 3.060 |
| Asthmatic HBEC | 14 d | 4.472 |
| | 20 d | 5.389 |
| | 6 d | 4.798 |
| Non-asthmatic HBEC | 14 d | 7.358 |
| | 20 d | 9.602 |

function in different experiments and simulations (*Atia et al., 2018*; *Kim et al., 2020*; *Li and Ciamarra, 2018*; *Lin et al., 2020*; *Wenzel and Voigt, 2021*). Nevertheless, since the deviation in *Figure 1d* is very weak, the values of $k$ are very close to each other. Therefore, the PDFs for $r_s$ in diverse epithelial systems–in experiments, simulations, and theory–should be nearly universal. To test this prediction, we have collected existing experimental and simulation data on different systems and show the PDFs of $r_s$ in *Figure 4c*. The variety in our chosen set is spectacular: it consists of various cancer cell lines (*Kim et al., 2020*), both asthmatic and non-asthmatic HBEC cells, MDCK cells (*Atia et al., 2018*), *Drosophila* wing disk (*Lin et al., 2020*), simulations data on both active (*Li and Ciamarra, 2018*) and equilibrium versions of the VM, the active Voronoi model (*Atia et al., 2018*), and the CPM. Yet, the PDFs shown in *Figure 4c* look nearly universal and in agreement with our analytical theory.

Additionally, our theory predicts a strictly universal behavior for $sd$ vs $\bar{r}$. Since this relation does not have any system-specific details, data across diverse confluent monolayers must follow a universal relationship. We have collected existing experimental data for several systems: cancerous cell lines (*Kim et al., 2020*), human breast cancer cells (*Ilina et al., 2020*), and a jammed epithelial monolayer of MDCK cells (*Fujii et al., 2019*). *Figure 4d* shows the experimental data together with our theoretical prediction; the agreement with our theory, along with the aspect of universality, is truly remarkable. As α increases, dynamics slows down, and the points on this plot move towards lower $\bar{r}$. This result is consistent with the finding that cell shapes are more elongated and variable as the dynamics become faster in different epithelial systems (*Atia et al., 2018*; *Park et al., 2015*).

We have argued that simultaneous measurements of the PDFs of cell area and $r$ distinguish the effects of maturation on the two key parameters: $\lambda_P$ and $T$. The argument relies on the negligible effect of the perimeter constraint on μ (*Equation 4*). We now show a comparison of our theoretical result for the PDF of $a$ with existing experiments. *Figure 4e* shows experimental data for four different systems (*Wilk et al., 2014*; *Puliafito, 2017*) and the corresponding fits of *Equation 4*. Unlike what has been proposed elsewhere that epithelial monolayers have a universal area distribution (*Wilk et al., 2014*), we find, in agreement with experiments, that although the functional form remain the same, the distribution can vary.

What are the implications of these universal aspects of cell shape variability and our theory? Cell shape controls several crucial biological functions such as the mitotic-orientation (*Wyatt et al., 2015*; *Bosveld et al., 2016*; *Xiong et al., 2014*) and cell fate (*McBeath et al., 2004*; *Wang et al., 2011*; *Roskelley et al., 1994*). Our theory shows that the microscopic system properties are encoded via a single parameter, $\alpha$. Consequently, knowledge of one of the observables, such as $\bar{r}$, contains the information of the entire statistical properties in a monolayer. We now illustrate this predictive aspect of the theory. Experimental measurement of an average property is usually less complex and more reliable. We have collected the data for $\bar{r}$ from the supplementary material of *Kim et al., 2020* for three different systems: 10Ca1A, 10AT, and 10 A.ErbB2, shown by the hexagrams in *Figure 4d*. From these average values, we obtain α, which we use to theoretically calculate the PDFs for $r$, as shown in *Figure 4f*. The inset of *Figure 4f* shows our theoretical PDFs for $r_s$, together with the corresponding experimental data for comparison. The excellent agreement demonstrates that cell shape variability results from the geometric constraint imposed by the energy function, *Equation 1*, and is not a choice but inevitable for such systems. This result, we believe, will foster analysis of diverse epithelial systems to understand the interrelation between geometric properties and biological functions within a unified framework.

## Discussion

We have obtained a mean-field theory for cell shape variability through the energy function $\mathcal{H}$, (*Equation 1*; *Farhadifar et al., 2007*; *Honda, 1978*; *Bi et al., 2015*; *Bi et al., 2016*; *Sadhukhan and Nandi, 2021*). We have used simplifying assumptions for analytical tractability and justified them in detailed simulations of the VM and the CPM on a square and a hexagonal lattice. The geometric restriction of confluency is a strong constraint on cell area. Considering that the area constraint is satisfied and that the cell cortex, described by the perimeter term, is crucial in determining the cell shape, allowed us to ignore the area term and obtain the distribution of $r$. We have justified this assumption in our simulations in the regime of our interest where cells are compact objects. A detailed comparison of our simplified analytical theory with simulations and experiments shows excellent agreements. Recent experiments and simulations have revealed that cell shape variability is nearly universal in confluent epithelial monolayers *Atia et al., 2018*; *Wenzel and Voigt, 2021*; *Kim et al., 2020*; *Ilina et al., 2020*; our work provides the theoretical basis for such behavior. We have shown that the universal properties are associated with a mathematical property and valid in general, even in the fluid regime; this is significant since most biological systems are in the fluid regime due to activity. Our analytical theory reveals that the microscopic system properties enter the distribution via a single parameter, α: this leads to the universal behavior for $sd$ vs $\bar{r}$ and a nearly universal distribution for the scaled aspect ratio $r_s$. Thus far, the PDF of $r_s$ has been fitted with a k-Gamma function with $k$ being around 2.5. We show that a rough approximation of our expression leads to the k-Gamma function; the slight variation in $k$ comes from the fact that the PDF is not strictly but nearly universal. On the other hand, $k \simeq 2.5$ is a direct consequence of a mathematical property: the lowest degeneracy of the eigenvalues of the connectivity matrix being two for a closed-looped object, here the perimeter.

A better understanding of the connection between the theoretical parameters and different system properties is crucial to exploit the universal aspects for deeper insights. Since all of the parameters combine into α, the effects of changing physical conditions on the individual parameters are difficult to determine from the measurements of $r$ alone. $\lambda_P$ describes the cortical properties, and $T$ parameterizes different biological activities, including temperature. These are effective parameters, and their direct estimation in biological systems is impractical. Our theory provides an indirect way to estimate these parameters. The cell area in a confluent system is geometrically constrained. We have used the phenomenological implementation of the constraint of confluency and obtained the PDF for $a$ (*Weaire, 1986*). It is a Gamma function, described by a single parameter μ. (*Wilk et al., 2014*) has proposed that the PDF of $a$ is universal in various epithelial monolayers. However, we show that though the area distribution follows the same function, there is a variation in μ. Our work connects μ to the microscopic model parameters of *Equation 1*. In particular, μ should be independent of $\lambda_P$, whereas α varies linearly with both $\lambda_P$ and $1/T$. This distinction, assuming $\lambda_A$ remains constant, allows inferring the effects of maturation on the individual model parameters.

We have neglected cell division, growth, and apoptosis in our theory. These are nonequilibrium processes, and the assumption of effective equilibrium becomes questionable. It will be interesting to see if the nonequilibrium statistical mechanics approach of *Grossman and Joanny, 2022* can be applied to obtain the distribution in the presence of these processes. However, more generally, since the rate of these processes are low, we expect the basic form of the distribution of $r$, *Equation 3* to remain the same. The algebraic part of *Equation 3* comes from the geometric property that remains the same. Therefore, we hypothesized that the effect of these additional processes should enter the distribution via α. We have included these processes in our simulations in Appendix 4. The results suggest that the general form of the distribution, and the prediction that all system-specific details enter via the single parameter α, remain valid when these processes are not dominant (Appendix 4).

Our work demonstrates that a single parameter, α, describes both the cell shape statistics and the dynamics. We have shown in our simulations that the relaxation time, $\tau$, grows as α increases or $\bar{r}$ decreases. Experiments on confluent cellular monolayer have also reported similar results (*Atia et al., 2018*; *Park et al., 2015*; *Kim et al., 2020*). Thus, as cells become more compact and their shape variability reduces, the monolayer becomes more sluggish. This result may have far-reaching consequences. Most experiments usually measure $\bar{r}$ and analyze biological functions via $\bar{r}$ (*Wyatt et al., 2015*; *Bosveld et al., 2016*). Our theory implies that such knowledge contains a wealth of information. One can obtain α from $\bar{r}$, and all other properties, such as the distribution, the standard deviation, and the dynamics, can be analytically calculated. Pattern formation in a biological system is

omnipresent during development. Pattern formation is generally controlled by gradients of different morphogens, such as the wingless or Dpp in the Drosophila wing (*Jaiswal et al., 2006*). Such gradients can be incorporated within our model. Since biological functions are related to cell shape, our results provide a statistical way to describe the system from the perspective of cellular functions, such as division or apoptosis. Such a description, in turn, will allow us to study the pattern formation in the presence of morphogen gradients. We are currently developing, in collaboration with colleagues, such a formalism to study the mechano-chemical pattern formation in the *Drosophila* wing disc.

It is well-recognized that different levels of organizations in biology are mechanistically related. One fundamental open question is how molecular-level events are related to cellular machines that control the cell shape (*Gilmour et al., 2017*). It is a difficult question as varying a specific property in a biological system is nearly impossible. Any perturbation will have significant impacts on several other proteins. Our work shows that all these perturbations enter the cell shape variability via a single parameter. The striking predictability, demonstrated by our theory, where $\bar{r}$ determines the PDF, shows that the statistical distribution of cell shape is unavoidable. How do different cells respond to this inevitable distribution? Is cellular response similar across diverse systems? How is it related to organ-level morphogenesis? Having a single parameter that describes the static and dynamic aspects at the cellular level should help compare and analyze different systems and answer these questions.

We have emphasized that the jamming and the glass transitions are distinct phenomena controlled by different physics (*Mari et al., 2009*; *Biroli and Garrahan, 2013*; *Berthier and Witten, 2009*; *Berthier et al., 2019*; *Atia et al., 2021*). The glass transition is not associated with any thermodynamic transition; therefore, we expect our results to remain valid even in the glassy regime. Although most of our simulations are in the fluid regime, some are also in the glassy regime. Glassiness is relevant only in the study of dynamics, and no static order parameter exists to date (*Berthier and Biroli, 2011*). Since our focus in this work is the static properties, we have not discussed the glass transition. By contrast, the shape index, $q$, has been shown to be an order parameter of the jamming transition (*Bi et al., 2015*; *Merkel et al., 2019*). Our formalism can also provide the distribution of $q$. These results will be presented elsewhere. In addition, our expression for the distribution of area is the same as that obtained for the Voronoi tessellated volume of a jammed granular system (*Aste and Di Matteo, 2008*). Although the Boltzmann distribution aspect at an effective temperature is similar, the bases of the two theories are quite different. This connection provides an alternative way to think about the constraint of confluency.

In conclusion, we have developed a simple mean-field theory for the aspect ratio distribution in a confluent epithelial monolayer. We show that the universal properties of a biological system, whose physics is controlled by the energy function *Equation 1*, come from a mathematical property. We have analytically derived the cell shape variability, characterized via $r$. The PDF of $r$ is described by a single parameter, $\alpha$. As a result, $sd$ vs $\bar{r}$ becomes universal, and the PDF for the scaled aspect ratio, $r_s$, is nearly universal. A rough approximation of our analytical form for the PDF of $r_s$ leads to the k-Gamma distribution (*Aste and Di Matteo, 2008*) that has been fitted to date with the existing experimental data (*Atia et al., 2018*; *Kim et al., 2020*). The distribution is valid in general, even in the fluid regime. The near-universal value of $k \sim 2.5$ is the consequence of a mathematical property; the variation results from the fact that the PDFs are not strictly universal. $\alpha$ can also provide information on dynamics. Having a single parameter for the statistical and dynamical aspects of an epithelial monolayer should foster a detailed comparison of diverse epithelial systems, provide insights on the relation of biological functions to shapes, and elucidate the detailed cellular responses to the inevitable shape variability.

# Materials and methods
## Details of the analytical calculation
To set up the calculation, we represent the cell perimeter in a coarse-grained description via $n$ points, where $l_j$ is the infinitesimal line-element between jth and (j+1)th points (see Appendix 5). Note that our final results are independent of this discretization of the perimeter. To calculate the aspect ratio, $r$, we first obtain the gyration tensor related to the moment of inertia tensor, a $2 \times 2$ tensor in spatial dimension two, in a coordinate system whose origin coincides with the center of mass (CoM) of the cell. Diagonalization of this tensor gives the two principal eigenvalues, $s_1^2$ and $s_2^2$, the squared-radii of

gyrations around the respective principal axes, and $r = s_1/s_2$. However, as discussed in the main text, a direct calculation of $s_1$ and $s_2$ is nontrivial due to their anisotropic natures. Therefore, we calculate the distribution of the radius of gyration, $s$, around the center of mass, and we have $s^2 = s_1^2 + s_2^2$ (**Davis and Denton, 2018**). The distribution of $s^2$ is

$$P(s^2) = \frac{1}{Z} \int \prod_{\rho=1}^{2} \delta(\sum_{j=1}^{n} x_j^\rho) \delta(1 - \frac{1}{ns^2}\mathbf{x}\mathbf{x}')e^{-\mathcal{H}_P/k_B T}\frac{\dot{x}}{ds}, \tag{5}$$

where $Z$ is the partition function, $\dot{x} = \prod_{\rho=1}^{2}\prod_{j=1}^{n} dx_j^\rho$, the volume element, $k_B$, the Boltzmann constant, and $T$, the temperature. The first $\delta$-function in **Equation 5** ensures that the origin coincides with the CoM of the cell; the second $\delta$-function selects specific values of $s^2$, giving the distribution function.

A precise mathematical description of adhesion remains unclear (**Graner and Riveline, 2017**; **Hilgenfeldt et al., 2008**; **Käfer et al., 2007**). The VM represents it via a line tension: $\sum_{\langle ij \rangle} \Lambda_{ij}\ell_{ij}$, where $\ell_{ij}$ is the length between two consecutive vertices, $i$ and $j$, and $\Lambda_{ij}$ gives the line tension. Since the degrees of freedom in the VM are the vertices, considering $\Lambda_{ij}$ constant, we obtain **Equation 1**. The constant $\Lambda_{ij}$ implies a regular cell perimeter between vertices. However, the entire cell boundary is in contact with other cells (**Figure 1a**), and the perimeter is often irregular in experiments (**Käfer et al., 2007**; **Atia et al., 2018**). Therefore, we take a more general description, where the tension in a line-element $l_j$ is proportional to $l_j$ with strength $P_0$. Thus, the adhesion part becomes $-\lambda_P\tilde{K}P_0\sum_j l_j^2$, where $\tilde{K}$ is a constant. At the cellular level description, this microscopic difference can be accounted for by a renormalized $P_0$. Since it is unclear how to measure $P_0$ in experiments, we mainly restrict our discussions on $\lambda_P$ and $T$. Unless otherwise stated, we assume $P_0$ remains constant. However, the theory does capture the variation in $P_0$, as we show within both the VM and the CPM simulations (**Figure 3h**).

Within our coarse-grained description, the contractile term is $P_i^2 = (\sum_j l_j)^2$. Now we can evaluate the integral, **Equation 5**, via a Gaussian approximation. However, an exact calculation, even at this level, is complicated. For analytical tractability and to gain insights, we use the Cauchy-Schwartz inequality (**Arfken et al., 2018**) and write $(\sum_j l_j)^2 \leq n\sum_j l_j^2 = \nu\sum_j l_j^2$, where $\nu \sim \mathcal{O}(n)$ is a constant. Note that this inequality is exact for the CPM, where we can consider $l_j = 1$ as the line element and $\nu = n$, the number of sides comprising the perimeter (see Appendix 5). In the experiments, the perimeter is usually obtained via a similar discretization (**Atia et al., 2018**). As shown in Appendix 5, this inequality, with the largest $n$, is a reasonable approximation for the perimeter term since the variation in perimeter is not too strong due to the perimeter constraint in **Equation 1**. Use of this inequality makes the evaluation of the integral slightly easier. In any case, it only affects the constant in the exponential that we treat as a fitting parameter within our theory. Since we are not investigating any transition here, the use of this relation is justified. Crucially, as we describe below, the parameter $k \simeq 2.5$ has a different origin that is not affected by this inequality. Then, the perimeter part of $\mathcal{H}$ becomes $\nu\tilde{\lambda}_P\sum_j l_j^2 = \nu\lambda_P(1 - KP_0)\sum_j l_j^2$, where $K = \tilde{K}/\nu$. The contractility and adhesion act as two competing effects (**Figure 1a**).

Thus, in the regime of our interest, the cell perimeter with the energy given by **Equation 2** governs the distribution of $r$ that we calculate via that of $s$. In the field of polymer physics, the distribution of $s$ has been calculated (**Fixman, 1962**; **Eichinger, 1977**; **Eichinger, 1980**). The mathematical structure of the two problems at this stage becomes equivalent. However, note that their physics are quite different. Specifically, our assumptions will not hold in the regime of interest of the polymer physics problem. To take advantage of an established notation and provide a connection between two disparate fields, we present our calculation in the notation of **Eichinger, 1977**; **Eichinger, 1980**. As discussed earlier, we describe the cell perimeter by the vector $\mathbf{x} = \{x_1^1, x_1^2, x_2^1, x_2^2, \ldots x_n^1, x_n^2\}$, representing a set of $n$ points on the perimeter. Then, we have $\mathcal{H}_P = \nu\tilde{\lambda}_P\mathbf{x}(\mathbf{K} \otimes \mathbf{I}_2)\mathbf{x}'$, where $\mathbf{x}'$ is the column vector, the transpose of $\mathbf{x}$, $\mathbf{K}$, the Kirchhoff's matrix (**Eichinger, 1977**; **Eichinger, 1980**) with $\mathbf{K}_{ii} = 2$ and $\mathbf{K}_{(i-1)i} = \mathbf{K}_{i(i-1)} = -1$, and $\mathbf{I}_2$, the two-dimensional identity tensor. Thus, we have

$$\mathcal{H}_P/k_B T = \gamma\mathbf{x}(\mathbf{K} \otimes \mathbf{I}_2)\mathbf{x}', \tag{6}$$

where $\gamma = \nu\lambda_P(1 - KP_0)/k_B T$. The distribution of $s^2$ is

$$P(s^2) = \frac{1}{Z} \int \prod_{\rho=1}^{2} \delta(\sum_{j=1}^{n} x_j^\rho) \delta(1 - \frac{1}{ns^2} \mathbf{x}\mathbf{x}') e^{-\gamma \mathbf{x}(\mathbf{K}\otimes\mathbf{I}_2)\mathbf{x}'} \frac{\dot{x}}{ds},$$

where the squared radius of gyration is $s^2 = n^{-1}\mathbf{x}\mathbf{x}'$. Since the radius of gyration does not depend on the coordinate system, we are allowed to chose one that diagonalizes $\mathbf{K}$. Say the diagonal matrix is $\mathbf{\Lambda}$, and $\mathbf{q}$ represents the normal coordinates in this system.

The radius of gyration can be defined as the root-mean-square distance of different parts of a system either from its center of mass or around a given axis. We have designated the former as $s$, defined as

$$s = \sqrt{\frac{1}{N} \sum_{i=1}^{N} (\mathbf{x}_i - \mathbf{x}_{CM})^2}, \tag{7}$$

where $N$ is the total volume element in the system with coordinates $\mathbf{x}_i$, and $\mathbf{x}_{CM}$ is the center of mass (CoM) of the system. The other two radii of gyration can be defined around the two principal axes (since we are in spatial dimension two) passing through the CoM. We calculate these two radii of gyration by writing the inertia tensor in a coordinate system whose origin coincides with the CoM and diagonalizing the tensor. The eigenvalues $s_1^2$ and $s_2^2$ are the squared-radii of gyrations around the respective principal axes. Thus, the aspect ratio, $r$, is obtained as $r = s_1/s_2$, assuming $s_1 \geq s_2$. As discussed in the main text, due to the anisotropic nature of $s_1$ and $s_2$, a direct calculation for their distributions is more complex than that of $s$. So, we first calculate the distribution for $s^2$ and then, using this result, we obtain the distribution of $r$.

*Equation 5* can be written in the normal coordinate system as.

$$P(s^2) = \frac{1}{Z} \int \prod_{\rho=1}^{2} \delta(q_n^\rho) \delta(1 - \frac{1}{ns^2} \mathbf{q}\mathbf{q}') \exp(-\gamma \mathbf{q}(\mathbf{\Lambda} \otimes \mathbf{I}_2)\mathbf{q}') \frac{\dot{q}}{ds}, \tag{8}$$

where we have used $q_n^\rho \propto \sum x_j^\rho$ that corresponds to the zero-eigenvalue mode of the matrix. Integrating over $q_n^\rho$, we get rid of this zero-eigenvalue that gives translation. Thus,

$$P(s^2) = \frac{1}{Z} \int \delta(1 - \frac{1}{ns^2} \mathbf{q}_0\mathbf{q}_0') \exp(-\gamma \mathbf{q}_0(\mathbf{\Lambda}_0 \otimes \mathbf{I})\mathbf{q}_0') \frac{\dot{q}_0}{ds}, \tag{9}$$

where we have defined $\mathbf{q}_0$ as the $2(n-1)$ dimensional vector excluding the coordinates corresponding to the zero-eigenvalue. The normalization factor, $Z$, can be calculated exactly through the integration as

$$Z \equiv \int \exp(-\gamma \mathbf{q}_0(\mathbf{\Lambda}_0 \otimes \mathbf{I}_2)\mathbf{q}_0')\dot{q}_0 = \left(\frac{\pi}{\gamma}\right)^{(n-1)} |\mathbf{\Lambda}_0|^{-1}. \tag{10}$$

Note that the integration in the calculation of $P(s^2)$ is around the boundary of the cell; to separate out the radial part, we now write the volume element in polar coordinate $\mathbf{u}$. Then $\dot{q}_0 = n^{(n-1)} s^{2(n-1)-1} ds\dot{u}$ and $\mathbf{q}_0\mathbf{q}_0'/ns^2 = \mathbf{u}\mathbf{u}' = 1$. Thus, we obtain from *Equation 9*

$$P(s^2) = \mathcal{A} \int_{-\infty}^{\infty} d\beta \int e^{-i\beta} e^{-\gamma ns^2 \mathbf{u}[(\mathbf{\Lambda}_0 - \frac{i\beta}{n\gamma s^2}\mathbf{I}_{n-1})\otimes\mathbf{I}_2]\mathbf{u}'} \dot{u}, \tag{11}$$

with $\mathcal{A} = \left(\frac{\gamma}{\pi}\right)^{(n-1)} |\mathbf{\Lambda}_0| \frac{1}{2\pi} n^{(n-1)} s^{2(n-1)-1} I_{n-1}$ is the identity matrix of rank $n-1$. Carrying out the integration over $\mathbf{u}$, we obtain

$$P(s^2) = \mathcal{A} \int_{-\infty}^{\infty} d\beta e^{-i\beta} \frac{\left(\frac{\pi}{\gamma ns^2}\right)^{(n-1)}}{\left|\mathbf{\Lambda}_0 - \frac{i\beta}{\gamma ns^2}\mathbf{I}\right|}. \tag{12}$$

Using the value of $\mathcal{A}$, we obtain

$$P(s^2) = \frac{1}{2\pi s} \int_{-\infty}^{\infty} d\beta \frac{e^{-i\beta}}{\prod_{j=1}^{n-1} \left(1 - \frac{i\beta}{\gamma n s^2 \lambda_j}\right)} = \frac{(\gamma n s^2)^{n-1}}{2\pi s} |\Lambda_0| \int_{-\infty}^{\infty} d\beta \frac{e^{-i\beta}}{\prod_{j=1}^{n-1} \left(\gamma n s^2 \lambda_j - i\beta\right)}, \tag{13}$$

where $\lambda_j$'s are the eigenvalues of $\mathbf{K}$. The integral in *Equation 13* can be performed via the contour integral and the resultant solution can be written as.

$$P(s^2) = \frac{(\gamma n s^2)^{n-1}}{2\pi s} |\Lambda_0| 2\pi i \sum_k Res(\lambda_k), \tag{14}$$

where $\lambda_k$ are the distinct eigenvalues of $\mathbf{K}$ and $Res(\lambda_k)$ gives the residue at the pole $\lambda_k$. As we show below, the residues will have a term $\exp[-n\gamma s^2 \lambda_k]$ and in the limit $s^2 \to \infty$, only the smallest $\lambda_k$ will contribute.

Since the cell perimeter must be closed-looped, $\mathbf{K}$ is a tridiagonal matrix with periodicity. Therefore, the number of zero-eigenvalues must be one, and the lowest degeneracy of the non-zero eigenvalues must be two (*Kulkarni et al., 1999*; *Witt et al., 2009*; *Eichinger, 1977*; *Eichinger, 1980*). We have already integrated out the coordinate corresponding to the zero-eigenvalue. Let us designate the lowest non-zero eigenvalue as $\lambda$. The pole corresponding to $\lambda$ is located at $\beta = -i\gamma n s^2 \lambda$, and of order 2. Thus, we obtain the residue as

$$Res = \frac{d}{d\beta} \left[ \frac{e^{-i\beta}}{\prod_{j=1}^{n-3} \left(\gamma n s^2 \lambda_j - i\beta\right)} \right]_{\beta = -i\gamma n s^2 \lambda}. \tag{15}$$

Let's first take the derivative, with respect to $\beta$, of the numerator and write part of the residue as

$$term1 = -i \frac{e^{-\gamma n s^2 \lambda}}{(\gamma n s^2)^{n-3} \prod_{j=1}^{n-3} (\lambda_j - \lambda)}. \tag{16}$$

Next, differentiating the denominator, we obtain the other part of the residue as

$$
\begin{aligned}
term2 &= e^{-i\beta} \left[ \frac{i}{(\gamma n s^2 \lambda_1 - i\beta)^2 \prod_{j=2}^{n-3} (\gamma n s^2 \lambda_j - i\beta)} + \frac{i}{(\gamma n s^2 \lambda_2 - i\beta)^2 \prod_{j=1, j\neq 2}^{n-3} (\gamma n s^2 \lambda_j - i\beta)} + \right. \\
&\left. \cdots \right]\Bigg|_{\beta = -i\gamma n s^2 \lambda} \\
&= i \frac{e^{-\gamma n s^2 \lambda}}{(\gamma n s^2)^{n-2} \prod_{j=1}^{n-3} (\lambda_j - \lambda)} \left[ \frac{1}{\lambda_1 - \lambda} + \frac{1}{\lambda_2 - \lambda} + \frac{1}{\lambda_3 - \lambda} + \cdots \right].
\end{aligned}
\tag{17}
$$

A comparison of term1 and term2, given by *Equation 16* and *Equation 17*, respectively, shows that there is an extra factor of $s^2$ in the denominator of term2. Thus, in the limit $s^2 \to \infty$ term2 can be ignored compared to term1. Therefore, we obtain the distribution function for $s^2$ as.

$$P(s^2) = \frac{|\Lambda_0| n^2 \gamma^2}{\prod_{j=1}^{n-3} (\lambda_j - \lambda)} s^3 e^{-\gamma n \lambda s^2} \equiv C s^3 e^{-\tilde{\alpha} s^2}, \tag{18}$$

where $C$ is the normalization constant that we will fix later, and $\tilde{\alpha} = \gamma n \lambda$. Note that the lowest eigenvalue for the $n$-dimensional Kirchoff's matrix is proportional to $1/n$, thus $n\lambda \sim \mathcal{O}(1)$.

Now, $s^2 = s_1^2 + s_2^2$ and the aspect ratio $r = s_1/s_2$. Moreover, we have $s_1 s_2 = A$, where $A$ is the area of the cell. Since the distribution of cell area is sharply peaked (*Figure 2c* in the main text), as the cell division and apoptosis are slow processes, $A$ can be taken as a constant. Therefore, using the last two relations in the first, we obtain $s^2 = A(r + 1/r)$. Thus, we obtain from *Equation 18*, the distribution of $r$ as.

$$P(r) = \frac{1}{\mathcal{N}} \left(r + \frac{1}{r}\right)^{3/2} \left(1 - \frac{1}{r^2}\right) e^{-\alpha(r + \frac{1}{r})}, \tag{19}$$

where $\mathcal{N}$ is the normalization constant: $\mathcal{N} = \Gamma(5/2)/\alpha^{5/2} - W(5/2){}_1F_1(5/2, 7/2, -2\alpha)$, where $W(x) = 2^x/x$, $\Gamma(x)$ is the Gamma function, and ${}_1F_1(a, b, c)$ is the Kummer's confluent Hypergeometric function (**Erdelyi et al., 1953**), $\alpha \propto \lambda_P(1 - KP_0)/T$.

Note that the exponent 3 in the algebraic part of **Equation 18** comes from the mathematical property, therefore, must be true for any system. This is the source of the value $k \simeq 2.5$ when **Equation 19** is approximated as a $k$-Gamma distribution, as discussed in the main text. For the analysis of the theory, a symbolic software, such as 'Mathematica' (**Wolfram Research Inc, 2019**) is helpful. We note here that the relation between $sd$ and $\bar{r}$ is quite complex, though for most practical purposes it can be taken as a straight line: $sd \simeq 0.71\,\bar{r} - 0.75$.

## Radius of gyration

There are different ways to define the radius of gyration depending on the application. Mathematically, it is the root mean square distance from the centre of mass or a given axis. We have designated the first as $s$ (**Equation 7**). For a two-dimensional object, there are two independent axes, therefore, there will be two radii of gyration, $s_1$ and $s_2$, in the second method. As detailed in the Discussion, $s_1$ and $s_2$ are obtained as the two eigenvalues of the gyration tensor. The significance of these two eigenvalues are as follows: the geometric properties of the system are equivalent to those of an ellipse with $s_1$ and $s_2$ as the major and minor axes. Considering that the mass is uniformly distributed throughout the system, the area is proportional to $s_1 s_2$ or $s^2$. We have used this property to obtain the distribution of $r$ in the analytical derivation of **Equation 19**.

## Distribution for area

**Equation 18** gives the distribution of the squared radius of gyration of different cells in a monolayer. Using this equation, we now derive the distribution of area, $A$. Note that **Equation 18** is valid irrespective of whether the system is confluent or not. We have argued in the main text that the constraint of confluency is not crucial to obtain the distribution of the aspect ratio. But, this argument is not valid when we are interested in the distribution of cellular area, $A$. Since confluency is a strong geometric constraint on $A$, we must include this constraint in the derivation of $P(A)$. Now, as detailed in 'Radius of gyration', $s^2 \propto A$. Therefore, **Equation 18** gives.

$$P(A) \sim A^{3/2} \exp[-\beta A], \tag{20}$$

where $\beta$ is a constant related to $\tilde{\alpha}$. We now consider the constraint of confluency. It is actually a long-standing hard mathematical problem. Even for random patterns, no exact result exists yet. (**Weaire, 1986**) proposed a phenomenological implementation of this constraint as a polynomial function of area; this remains one of the simplest possible ways to date to deal with this constraint (**Gezer et al., 2021**). Keeping only one term of this polynomial for simplicity, we can write this constraint as $f(A) \sim A^\nu$ (**Weaire, 1986**; **Gezer et al., 2021**). Using this in **Equation 20**, we obtain

$$P(A) \sim A^{\mu - 1} \exp[-\beta A], \tag{21}$$

where we have defined $\mu = \nu + 5/2$. From **Equation 21**, we obtain the average area, $\bar{A} = \mu/\beta$. Thus, we obtain the normalized distribution for the scaled area, $a = A/\bar{A}$, as

$$P(a) = \frac{\mu^\mu}{\Gamma(\mu)} a^{\mu - 1} \exp[-\mu a]. \tag{22}$$

This is the well-known k-Gamma function, defined in **Aste and Di Matteo, 2008**, and usually denoted with the variable $k$. This same function has been used in fitting the scaled aspect ratio, $r_s$, data in different existing experiments and simulations. Therefore, to avoid confusion with $k$, which is obtained fitting the $r_s$ data, we have used $\mu$ to define the distribution function for $a$. Since $\mu$ comes from the constraint of confluency, it should be independent of $\lambda_P$. Our simulation results within both the CPM and the VM (**Figure 3e**, in the main text) support this hypothesis.

## Acknowledgements

We thank Satyabrata Bandyopadhyay, Mustansir Barma, Tamal Das, Manish Jaiswal, Sritama Datta, Basil Thurakkal. and Sumedha for many enlightening discussions and Sam A Safran for comments. We acknowledge the support of the Department of Atomic Energy, Government of India, under Project Identification No. RTI 4007 and the computational facility of TIFR Hyderabad. We also thank SERB for grant via SRG/2021/002014.

## Additional information

### Funding

| Funder | Grant reference number | Author |
|---|---|---|
| Department of Atomic Energy, Government of India | RTI 4007 | Souvik Sadhukhan Saroj Kumar Nandi |
| Science and Engineering Research Board | SRG/2021/002014 | Saroj Kumar Nandi |

The funders had no role in study design, data collection and interpretation, or the decision to submit the work for publication.

### Author contributions

Souvik Sadhukhan, Software, Formal analysis, Investigation, Visualization, Methodology, Writing – original draft, Writing – review and editing; Saroj Kumar Nandi, Conceptualization, Formal analysis, Supervision, Funding acquisition, Investigation, Visualization, Writing – original draft, Writing – review and editing

### Author ORCIDs

Souvik Sadhukhan http://orcid.org/0000-0002-0926-6601
Saroj Kumar Nandi http://orcid.org/0000-0003-0977-1947

### Decision letter and Author response

Decision letter https://doi.org/10.7554/eLife.76406.sa1
Author response https://doi.org/10.7554/eLife.76406.sa2

## Additional files

### Supplementary files
• Transparent reporting form

### Data availability

We have uploaded the source files of the simulation data and the Mathematica analysis files in Dryad.

The following dataset was generated:

| Author(s) | Year | Dataset title | Dataset URL | Database and Identifier |
|---|---|---|---|---|
| Nandi S | 2022 | Data from: On the origin of universal cell shape variability in a confluent epithelial monolayer | http://dx.doi.org/10.5061/dryad.xsj3tx9h0 | Dryad Digital Repository, 10.5061/dryad.xsj3tx9h0 |

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

## Appendix 1

### Boltzmann Distribution

As we discussed in the main text, there are two types of models of confluent systems, depending on the presence or absence of self-propulsion. In the absence of self-propulsion, the models are in equilibrium. It has been shown in the literature, that the equilibrium variants of the models do capture the static and dynamical properties of a confluent epithelial system (*Farhadifar et al., 2007*; *Atia et al., 2018*; *Park et al., 2015*). For concreteness, we have considered the equilibrium variants of the models in this work. Therefore, we expect that the Boltzmann distribution should hold, that is, a configuration with energy $E$, given by *Equation 1*, should have the weightage of $\sim \exp[-E/k_B T]$. However, as discussed below in Appendix 2, we restrict ourselves in the low-$P_0$ regime, where the cells, on average, cannot satisfy the perimeter constraint. This implies a nontrivial dependence of the distribution around $E \to 0$ coming from the constraint of confluency that in turn restricts the minimum value of the perimeter; the energy cannot go below a certain value. However, we show in *Appendix 1—figure 1* that we can still approximate the distribution via a Boltzmann distribution.

If the system follows the Boltzmann distribution, that is, $P(E) \sim \exp(-E/T)$, plots of $\log[P(E)]$ as a function of $E/T$ for different $T$ should follow a master curve that is a linearly decreasing function. As shown in *Appendix 1—figure 1*, except at very close to $E \to 0$, $P(E)$ as a function of $E/T$ at different $T$ indeed follows a master curve, implying the existence of the Boltzmann distribution.

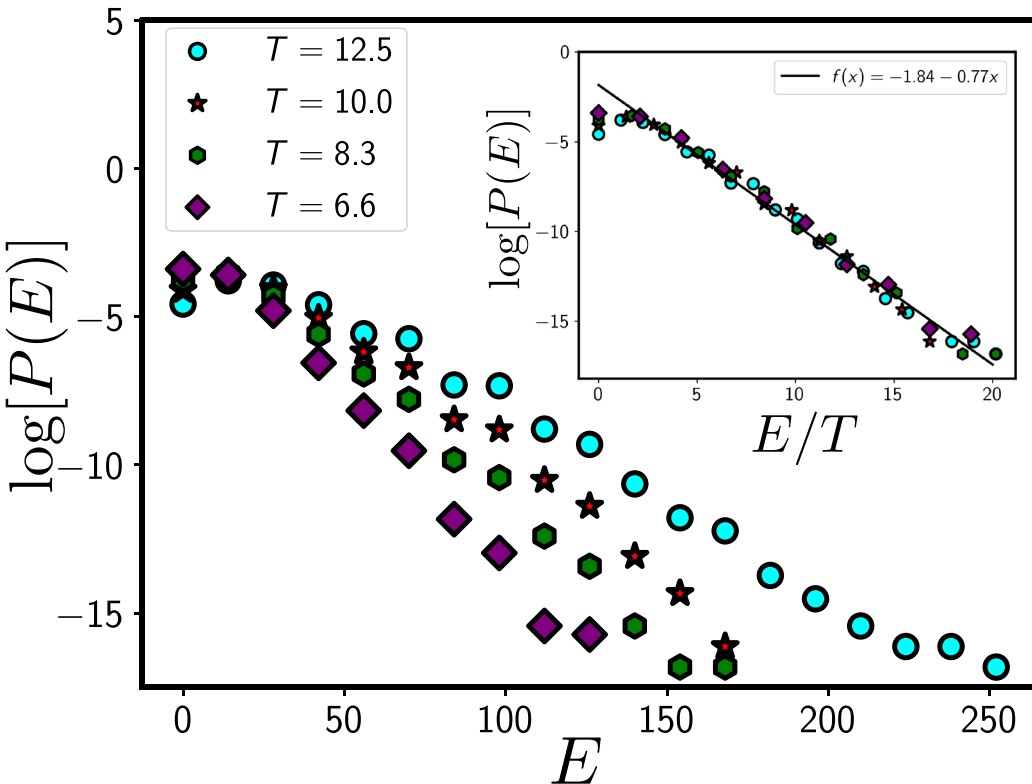

**Appendix 1—figure 1.** The systems in our simulation follow a Boltzmann distribution. If the system follows a Boltzmann distribution at a temperature T, plot of the logarithm of distribution log[P(E)] as a function of E/T at different T should follow a master curve that linearly decreases with E/T. In the main figure, we show the distribution, P/E at different T and the inset shows the data collapse for P/E as a function of E/T. The data presented in this figure are obtained from the CPM simulation of a total system size 120X120 with 360 cells. We have set $P_0 = 26$ and $A_0 = 40$. We have equilibrated the system for $10^5$ time steps before collecting data.

## Appendix 2

### Simulation details

We have verified our theory via simulations of two distinct models: the CPM and the VM. In both the simulations, we have used the energy function $\mathcal{H}$, given in **Equation 1** in the main text. The CPM is a lattice-based model. The underlying lattice has some effects on the quantitative aspects of the model; for example, on the square lattice, the object with the minimum possible perimeter for a given area is a square. However, the qualitative behaviors are independent of the lattice. To ascertain that the simulation results are independent of the lattice, we have simulated the CPM on two different lattices: the square lattice and the hexagonal lattice.

For the glassy dynamics, the geometric restriction leads to two different regimes, the low-$P_0$, and the large-$P_0$ regime. However, the large-$P_0$ regime characterizes quite large adhesion compared to the cortical contractility; this regime, we feel, is not relevant for the experiments. Therefore, we present most results when $P_0$ is not too large. Here, we briefly discuss the different models.

### Cellular Potts Model (CPM)

In CPM, dynamics proceeds by stochastically updating one boundary lattice site at a time. In a Monte-Carlo simulation (MC), we accept a move with a probability, $P(\sigma \rightarrow \sigma') = e^{-\frac{\Delta \mathcal{H}}{T}}$, where we have set Boltzmann constant $k_B$ to unity, $\Delta \mathcal{H}$ is the change in energy going from one configuration to the other. $\sigma$ and $\sigma'$ are the cell indices of the current cell and the target cell indices, respectively. Unit of time (1 MC time) refers to $M$ such elementary moves, where $M$ is the total number of lattice sites in the simulation.

### With square lattice

We have implemented CPM with square lattice in Fortran 90 and followed a Connectivity Algorithm developed in **Durand, 2016** to prevent fragmentation of cells (**Sadhukhan and Nandi, 2021**). We have chosen a simulation box of size $120 \times 120$ with 360 total cells in the system. The average area of cells in the system is 40, and the minimum possible perimeter on a square lattice with this area is 26. Unless otherwise specified, we always start with an initial configuration where each cell is rectangular with a size $5 \times 8$. We first equilibrate the system for $8 \times 10^5$ MC time before collecting the data of aspect ratio. The variables that characterize the system are $\lambda_A$, $\lambda_P$, $P_0$, and $T$. We have ignored cell division and apoptosis, as discussed in the main text. However, to test the effects of these processes, we have included them within this model and present a specific set of results in Appendix 4.

To obtain the relaxation time, $\tau$, we have calculated the overlap function, $Q(t)$, defined as

$$Q(t) = \overline{\frac{1}{N} \sum_{\sigma=1}^{N} \langle W(a - |\mathbf{X}_{cm}^{\sigma}(t + t_0) - \mathbf{X}_{cm}^{\sigma}(t_0)|) \rangle_{t_0}}, \quad (23)$$

where $X_{cm}^{\sigma}(t)$ is the center of mass at time $t$ of a cell with index $\sigma$, $W(x)$ is a Heaviside step function

$$W(x) = \begin{cases} 1 & \text{if } x \geq 0 \\ 0 & \text{if } x < 0, \end{cases} \quad (24)$$

$\langle \ldots \rangle_{t_0}$ denotes averaging over initial times and the overline implies ensemble average. Relaxation time is defined as $Q(t = \tau) = 0.3$. The parameter $a$ is related to the vibrational motion of the particles (here cells) inside the cage formed by its neighbors. We have set $a = 1.12$ (**Sadhukhan and Nandi, 2021**). We have taken 50 $t_0$ averaging and 20 configurations for ensemble averaging.

### With Hexagonal lattice

To simulate CPM on the hexagonal lattice, we have used an open-source application CompuCell3D (**Swat, 2012**; **Zajac et al., 2003**). CPM is the core of CompuCell3D. We have simulated the 2D confluent cell monolayer with periodic boundary conditions using the same energy function, **Equation 1**, in the main text. In this simulation, we have taken $128 \times 128$ simulation box size with 361 cells. Initially, Cells are of three different types of areas and perimeters. We have started with a rectangular slab of the cell monolayer. We first equilibrate the system for $10^5$ MC time steps before collecting the data. Fragmentation is allowed in these simulations; however, we have restricted

ourselves in the low $T$ regime, where fewer cells are fragmented. While calculating the PDFs, we have eliminated the fragmented cells.

## Vertex Model (VM)

Vertex models (VM) can be viewed as the continuum version of the CPM. Within VM, vertices of the polygons are the degrees of freedom, and cell edges are defined as straight lines (or lines with a constant curvature) connecting between vertices (*Farhadifar et al., 2007*; *Fletcher et al., 2014*). Within the Monte-Carlo simulatoin (*Wolff et al., 2019*), dynamics proceeds by stochastically updating each vertex position of all the cells by a small amount $\delta r$. We accept the move with a probability $P(\mathcal{C} \to \mathcal{C}') = e^{-\frac{\Delta \mathcal{H}}{T}}$ where we have set Boltzmann constant $k_B$ to unity, $\Delta \mathcal{H}$ is the change in energy going from $\mathcal{C} \to \mathcal{C}'$. Initially, we start will 1024 cells having equal area (= 1.0) and perimeter (= 3.72). We have equilibrated the system for $2 \times 10^6$ MC time before the start of collecting data.

## Two different regimes

As we discussed in the main text, theoretical studies have shown the existence of two distinct regimes as a function of $P_0$ within these models of confluent epithelial cells (*Sadhukhan and Nandi, 2021*; *Bi et al., 2015*). The existence of these two regimes is controlled by a geometric constraint. Given a certain average cell area, the perimeter can have a minimum value. If $P_0$ in *Equation 1* is less than this value, cells cannot achieve this target perimeter: the cells in this regime are compact. It seems that this is the regime of experimental interest. In contrast, when $P_0$ is large compared to this minimum value, cells can achieve this value to minimize energy $\mathcal{H}$. Within the CPM, the cells look like fractal objects in this regime (*Sadhukhan and Nandi, 2021*). This behavior is expected to be similar within both the VM and the Voronoi model. But there are technical difficulties to simulate these latter models in this regime due to numerical instabilities.

## Appendix 3

### Procedure to calculate aspect ratio ($_r$) in our simulation

As stated in the main text, we characterize the cell shape via aspect ratio, $r$. To calculate $r$, we follow (**Atia et al., 2018**) and briefly discuss the method here. Consider one particular cell, represented by the set of points $\{x_i^1, x_i^2\}$, as schematically shown in **Appendix 3—figure 1**. We first calculate the gyration tensor, $\mathbf{I}$, (proportional to the moment of inertia tensor) in a frame of reference whose origin coincides with the center of mass ($x_c^1, x_c^2$) of the cell. $\mathbf{I}$ is a $2 \times 2$ tensor in spatial dimension two. We next diagonalize $\mathbf{I}$ and obtain the two eigenvalues, $s_1^2$ and $s_2^2$. Without loss of generality, we assume that $s_1 \geq s_2$, and obtain $r = s_1/s_2$. This process can be viewed as approximating the cell with an ellipse with the major and minor axes given by $s_1$ and $s_2$, respectively. For the CPM on the square lattice and the VM, we have written our own codes; for the CompuCell3D (**Swat, 2012**; **Zajac et al., 2003**), we have used <Plugin Name="MomentOfInertia"/>in the.XML file and using the example of Demos/ MomentOfInertia, we have calculated the lengths of semiaxes in the Steppables.py file. We have obtained the aspect ratio of a cell by taking the ratio of lengths of semi-major and semi-minor axes.

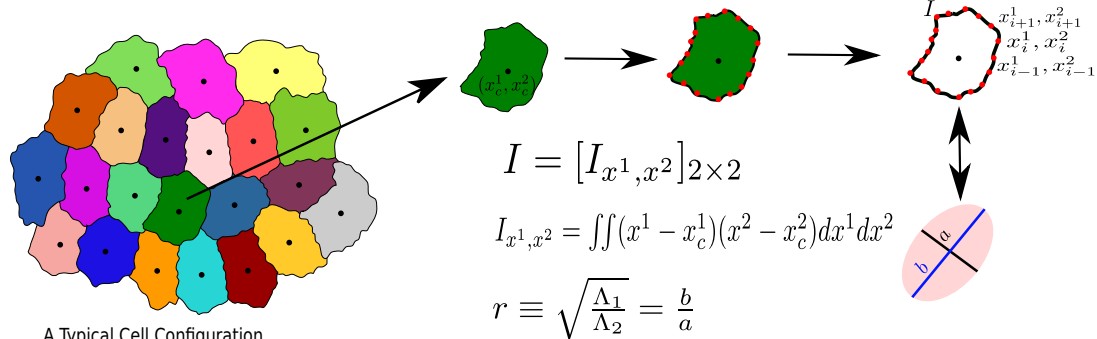

A Typical Cell Configuration

$$I = [I_{x^1, x^2}]_{2\times 2}$$

$$I_{x^1, x^2} = \iint (x^1 - x_c^1)(x^2 - x_c^2) dx^1 dx^2$$

$$r \equiv \sqrt{\frac{\Lambda_1}{\Lambda_2}} = \frac{b}{a}$$

**Appendix 3—figure 1.** The process to calculate the aspect ratio of a Cell. We first obtain the moment of inertia, I, in a frame of reference whose origin coincides with the center of mass of the cell, then diagonalize I, and calculate r, as the square root of the ratio of the two eigenvalues, $s^2_1$ and $s^2_2$, assuming s1 ≥ s2.

# Appendix 4

## Including cell division and apoptosis

We have ignored cell division and apoptosis in our theory since the rates of these processes are extremely low. However, even when these processes are present, as long as their rates are not very high, we expect the form of the distribution, *Equation 3* in the main text, should remain valid. There are two distinct parts in $P(r)$. The algebraic part is determined by the geometric property, which is a cell perimeter is a closed-loop object. This property cannot change. Therefore, the effects of these processes can enter the distribution via $\alpha$ alone. Thus, the main conclusions of the theory should remain valid even when division and apoptosis are present.

We have included cell division and apoptosis in our simulations to test this hypothesis. We keep the rate of division, $k_d^{-1}$, and apoptosis, $k_a^{-1}$, the same such that the average number of cells remains the same. Every $k_d$ time step, we randomly select a cell and divide it into two, with a randomly chosen division plane. To decide the division plane, we first calculate the center of mass (CoM) and then chose a point on the perimeter in a random direction. The line connecting the CoM and this point gives the division plane (a line for the two-dimensional system). The area of these two daughter cells then grows till it becomes of the order of $A_0$ (*Equation 1* in the main text). To avoid dividing a cell that has just undergone division, we impose a cut-off on the area ($\geq 32$) for selecting a cell for division. For apoptosis, every $k_a$ time step, we randomly choose a cell and assign it a target area $A_0 = 0$. We also relax the constraint that does not allow fragmentation in our simulation for the cells undergoing apoptosis.

We show the PDF of $r$ at three representative values of $k_a$ above. As shown in *Appendix 4—figure 1a* PDFs of r at different values of $k_d = k_a$ can be fitted with *Equation 3*, the values of α for the best fits are quoted in the figure. Crucially, an important prediction of the theory that the PDF for $r_s$ becomes nearly universal remains also valid, as shown in *Appendix 4—figure 1b*. We find that α as a function of $k_d$ or $k_a$ quickly reaches a constant with increasing $k_a$ (that is, with decreasing rate) (*Appendix 4—figure 1c*).

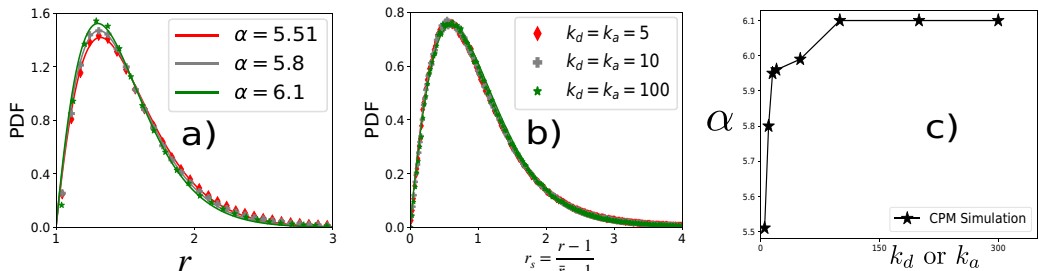

**Appendix 4—figure 1.** PDF of r at three representative values of $k_a$. (**a**) PDFs of r at different values of $k_d = k_a$ (quoted in **b**) within CPM for $\lambda_P = 0.5$ and T = 12.5. (**b**) PDF of $r_s$ corresponding to the same data as in (**a**). The lines are fits with *Equation 3* of the main text. (**c**) α vs $k_d$ or $k_a$ quickly reaches a constant with increasing $k_a$.

## Appendix 5

### Validity of the approximation of perimeter

We have used the Cauchy-Schwartz inequality, $(\sum_j l_j)^2 \leq n \sum_j l_j^2 = \nu \sum_j l_j^2$, where $\nu \sim \mathcal{O}(n)$ is a constant. Within the CPM, the perimeter consists of $n$ elementary sides, as schematically shown in **Appendix 5—figure 1a** above, then $P_i^2 = [\sum_{j=1}^n \ell_j]^2$. Since $\ell_j = 1$, $P_i^2 = [\sum_{j=1}^n \ell_j]^2 = n \sum_{j=1}^n \ell_j^2$. The inequality is exact in that case. The data from the simulation confirms this (**Appendix 5—figure 1**). In the case of the continuum models, such as the vertex model, we can use a similar discretization as schematically shown by the red line in the inset of **Appendix 5—figure 1**. In fact, the estimate of cell perimeter in the experiments is obtained via a similar discretization method. With such a discretization representing the continuous cell perimeter via a discrete line, we again find that the inequality is very close to equality (**Appendix 5—figure 1b**).

The approximation of replacing the inequality by equality now depends on the fluctuation of the total perimeter or $n$. This fluctuation depends on the values of the model parameters, such as $T$, $\lambda_P$, $P_0$, etc. We find that for the parameter values used in this work to simulate the systems, the highest fluctuation of the perimeter, or in the value of $n$, is less than 5%. Thus, we are justified to approximate the inequality by equality.

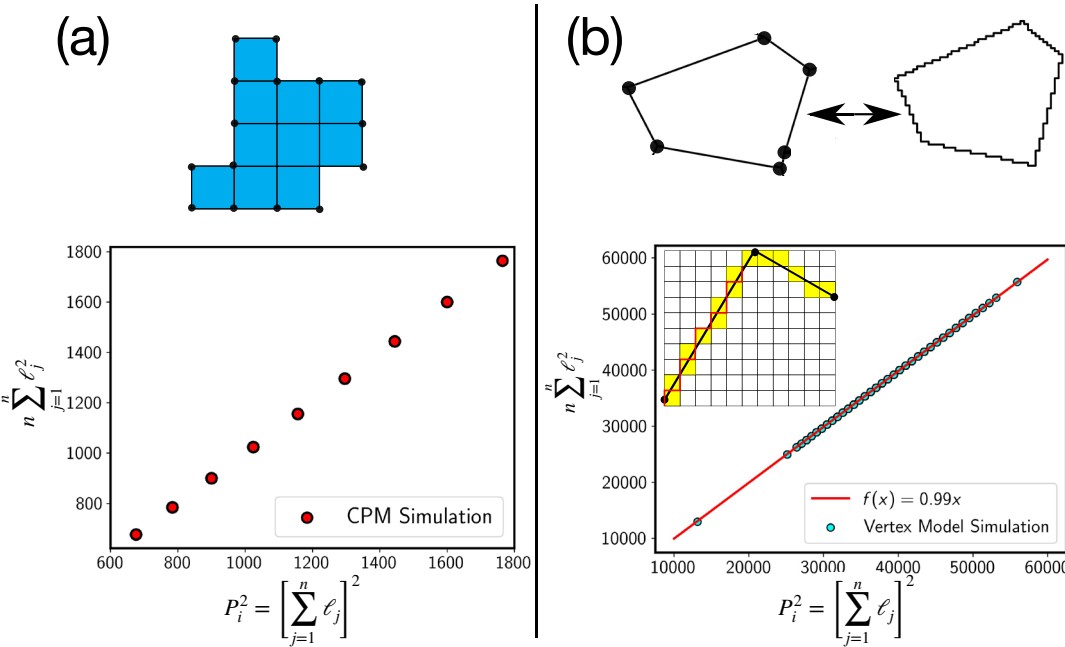

**Appendix 5—figure 1.** Test of the perimeter inequality. (**a**) In the CPM, the approximation of $P^2_i$ as $n \sum_j l_j^2$, where $n$ is the number of elementary lengths, $\ell_j$ is exact as can be seen from plot where the two estimates are equal. (**b**) For the continuum models, such as the vertex model, we can use a similar discretization as schematically shown in the figure. Experimental estimate of the perimeter comes from such a discretization, where the continuous cell perimeter is represented by a discrete line, as schematically shown with the red line. Even in this case, we find that the inequality is very close to an equality.

