## [Editor Report]

In this important study, the authors unveil the reason for nearly universal shape fluctuations that have been reported earlier by theoretically analysing the energy of a confluent epithelial tissue. The comparison of their analytic results with simulations and experimental data is compelling, only the justification of the cell area distribution is somewhat incomplete. The manuscript is relevant for all people with an interest in tissue structure and dynamics.

---

## [Decision Letter]

**Decision letter after peer review:**

Thank you for submitting your article "On the origin of universal cell shape variability in a confluent epithelial monolayer" for consideration by *eLife*. Your article has been reviewed by 2 peer reviewers, one of whom is a member of our Board of Reviewing Editors, and the evaluation has been overseen by a Reviewing Editor and Aleksandra Walczak as the Senior Editor. The following individual involved in review of your submission has agreed to reveal their identity: Jean-François Joanny (Reviewer #2).

Essential revisions:

1) The authors consider a regime where the area is equal to the target or preferred area of the cells and the area part of the free energy can be neglected. Quantitatively check the validity of this approximation.

2) Justify, why effectively equilibrium statistics can be used in the analysis.

3) In Appendix A you replace the square of the perimeter by the sum of the squares of the lengths of the edges of the cell. Check the accuracy of this approximation quantitatively.

4) Make the text more accessible to biologists.

*Reviewer #1 (Recommendations for the authors):*

I do not have any recommendations for further analysis or computation. However, the way the manuscript is written is rather poor. The authors often use jargon and, at places, the text is repetitive and confusing. Below I give some examples that might help the authors to make their work more accessible to the broad readership of *eLife*.

The authors very often switch between AR and r for denoting the aspect ratio. The figures sometimes, but not always contain both notions. I would strongly suggest introducing one symbol only and using it throughout the manuscript.

2nd paragraph is very long and partly repetitive.

p.2

"In a jammed granular system, the statistics of tessellated volume, x, follows a k-Γ distribution," This sentence is not very biologist friendly.

"Almost universal" Other expressions used are "nearly universal", "virtually universal". Neither of them is well-defined and it is not clear if they all refer to the same feature. If yes, use only one notion. Also, try to make it scientific and not just intuitive.

"The existing theoretical works on cellular shapes are based on the elasticity framework." Again, hard to understand for a biologist.

p.3

On this page there a several jumps in the presentation. Try to streamline.

"One crucial aspect of these systems" Which ones?

"the cortex, a thin layer of cytoplasm" Better: the cortex is a thin actomyosin layer

"we mostly focus our discussions below within the CPM " but then "cells are compact objects (in contrast to being fractal-like in the CPM simulation [30])" Can you clarify the compact vs fractal nature of the cells in the CPM you use?

"we first need to obtain the two radii of gyrations, s1 and s2, around the two principal axes." You should probably explain 'radius of gyration' for biologists as well as their relation to cell area.

"as detailed in the Appendix B, Equation (??), presented in the Appendix," Number is missing.

"Since μ is related to the constraint of confluency" Here Ii was confused, because you consider an isolated cell, right?

p.4

"Note that the power 3/2 of the algebraic term in Equation (3) comes from the mathematical property of closed-looped objects." Again, this requires more explanation for biologists

Figure 1d you could maybe show a log-log plot.

p.5

"Figure 3g" It seems to be premature to refer to this figure here.

"Finally, we show" Where?

"although they follow the same distribution, they need not be identical" What are 'they'? Also, I find it hard to understand this sentence.

"we have simulated the confluent systems with varying λ_A and find that the AR distribution remains almost independent of λ_A" Can you show the distribution?

"as T changes, the PDF of a, though well-described by a single parameter μ via Equation (4), can still be different" Yo mean the values of \mu can be different?

Figure 4c-e Why do you not show all curves for all cell types?

Figure 4d: Hexagons? Maybe stars?

p.10

last paragraph of the main text is just a repetition of the previous paragraphs

*Reviewer #2 (Recommendations for the authors):*

– The authors consider a regime where the area is equal to the target or preferred area of the cells and the area part of the free energy can be neglected. Can the validity of this approximation be made more quantitative? A naive guess would be that the modulus \λ_a is very large but in the end the authors calculate fluctuations of the area due to shape fluctuations and they would cost a large energy which is not taken into account. In the vertex model, as explained by the authors there is a transition between a soft solid and a hard solid behavior and the transition only depends on the ratio between the perimeter and the square root of the area. Which regime do the authors consider?

– The authors define the tissue by an energy which is used in all the models that they quote. In my mind this energy is an efficient way to write that the forces in the epithelial layer are balanced. However, they then use a Boltzmann Boltzmann statistics with a well defined temperature for the configuration of the tissue. This temperature seems to play an important role as it enters the expression for the parameter \α. Is this equilibrium statistics also used in the simulations? In my mind, a tissue is an intrinsically non-equilibrium system and I do not see how the equilibrium distribution can be justified. The authors also claim that the results are not changed if cell division is considered, which is difficult for me to understand. The manuscript would be much stronger if the authors were discussing this effective equilibrium assumption in details.

– Another approximation is made in appendix A which replaces the square of the perimeter by the sum of the squares of the lengths of the edges of the cell. I understand that this strongly simplifies the calculations. Could the authors make more quantitative this approximation which is only justified by an inequality. How accurate is it?

– On more general grounds, the results obtained by the authors are interesting but do they have any biological relevance? Could the authors point out a biological phenomenon for which cell shape fluctuations would play an important role?

---

## [Author Response]

Essential revisions:Reviewer #1 (Recommendations for the authors):I do not have any recommendations for further analysis or computation. However, the way the manuscript is written is rather poor. The authors often use jargon and, at places, the text is repetitive and confusing. Below I give some examples that might help the authors to make their work more accessible to the broad readership of eLife.

We sincerely thank the reviewer for the insightful comments, appreciation of the importance of the work, and suggestions to make the manuscript accessible to a broader audience. We have revised the manuscript accordingly and removed the repetitions. We hope the reviewer and the editor finds the revised manuscript accessible to a broad audience. Here we briefly respond to the comments of the reviewer.

The authors very often switch between AR and r for denoting the aspect ratio. The figures sometimes, but not always contain both notions. I would strongly suggest introducing one symbol only and using it throughout the manuscript.

Thanks, we have now used the notation r to denote the aspect ratio.

2nd paragraph is very long and partly repetitive.

We have now shortened this paragraph and removed the repetitions.

p.2"In a jammed granular system, the statistics of tessellated volume, x, follows a k-Γ distribution," This sentence is not very biologist friendly.

We have expanded this sentence, explaining what are these volumes: “In a jammed system, one can carry out …. by a single parameter *k* [41].”

"Almost universal" Other expressions used are "nearly universal", "virtually universal". Neither of them is well-defined and it is not clear if they all refer to the same feature. If yes, use only one notion. Also, try to make it scientific and not just intuitive.

We thank the reviewer for this suggestion. They do refer to the same feature and we now use the term “nearly universal” alone. We have also explained what it means (Page 5, left column): “the *α*-dependence becomes so weak … independent of *α*.”

"The existing theoretical works on cellular shapes are based on the elasticity framework." Again, hard to understand for a biologist.

We have now rephrased this sentence explaining the elasticity framework: “The existing theoretical works … when external force is relaxed.” (Page 2, first column).

p.3On this page there a several jumps in the presentation. Try to streamline.

Thanks for pointing this out. We have now made it more contiguous by providing the details, either in the Appendix or within the main text itself. We have also streamlined the presentation by providing more details in the Materials and methods Section and including additional details in the Appendix.

"One crucial aspect of these systems" Which ones?

We have explicitly mentioned the models now: “One crucial aspect of these model systems (such as the VM, the CPM, or the Voronoi model) of epithelial monolayer is the constraint of confluency”.

"the cortex, a thin layer of cytoplasm" Better: the cortex is a thin actomyosin layer

We have rephrased it as “First, a thin actomyosin layer, known as cortex, mainly governs the cellular mechanical properties”.

"we mostly focus our discussions below within the CPM " but then "cells are compact objects (in contrast to being fractal-like in the CPM simulation [30])" Can you clarify the compact vs fractal nature of the cells in the CPM you use?

There are two regimes of these models, the system behaves like soft and hard solids in these two regimes. We mainly focussed on the CPM, because from the theoretical perspective, this model has a lot of advantage and easier to understand many of the complex behaviours. Within the CPM, the cells look like fractal objects and compact objects, respectively in these two regimes. We have added a subsection in Appendix C, titled “Two different regimes”, explaining this behaviour (page 15, last subsection of Appendix 2).

"we first need to obtain the two radii of gyrations, s1 and s2, around the two principal axes." You should probably explain 'radius of gyration' for biologists as well as their relation to cell area.

We have added a section, “Radius of gyration”, in the Materials and methods Section with the various definitions of radii of gyration and their relation with the area (Page 13, Section IV B).

"as detailed in the Appendix B, Equation (??), presented in the Appendix," Number is missing.

Thanks for pointing this out, we have fixed it in the revised manuscript.

"Since μ is related to the constraint of confluency" Here Ii was confused, because you consider an isolated cell, right?

We apologise for this confusion. Actually, there are two different calculations presented in this work: the distribution of aspect ratio and the distribution of the cell area. The first one is the main result of the work, but the second result is also important as it connects the derivation to several other problems.

Our interest is the cellular aspect ratio in a confluent monolayer. But the constraint of confluency is an extremely hard mathematical problem and no exact results exist till date to treat it analytically. Motivated by our simulation results, we have argued that the constraint of confluency is not crucial for the aspect ratio distribution. We have also verified this in our simulations. This slightly simplifies the derivation of *P*(*r*).

But, for the distribution of cellular area, the constraint of confluency is crucial. Equation

(18) is valid for a general system. We then included this constraint in our analytical derivation following Weaire et al. (Ref. 67) and obtained the distribution of cell area.

We have clarified this in the text in section Materials and methods, IV C.

p.4"Note that the power 3/2 of the algebraic term in Equation (3) comes from the mathematical property of closed-looped objects." Again, this requires more explanation for biologists.

We have detailed this point in the Appendix A. In the main text (page 4, second paragraph), we have now stated the property that leads to this value and has referred the reader to the appendix: “That is, for closed-loop objects, …. As detailed in Sec. IV A”.

Figure 1d you could maybe show a log-log plot.

We have now shown the plot in log-log scale in the inset of Figure 1(d). p.5

p.5"Figure 3g" It seems to be premature to refer to this figure here.

We have removed the reference to Figure 3(g) here.

"Finally, we show" Where?

Thanks, we have now added the reference [Figure 1(f)].

"although they follow the same distribution, they need not be identical" What are 'they'? Also, I find it hard to understand this sentence.

We have now clarified this sentence: “The PDFs of *a* has been argued… they are not identical”.

"we have simulated the confluent systems with varying λ_A and find that the AR distribution remains almost independent of λ_A" Can you show the distribution?

We have now included these distributions in the supplementary figure, Figure 2 —figure supplement 2 and added in the main text: “(the distributions are shown in Figure 2 —figure supplement 2).”

"as T changes, the PDF of a, though well-described by a single parameter μ via Equation (4), can still be different" Yo mean the values of \mu can be different?

Yes. We have now rephrased this sentence for better clarity: “… by the same function, Equation (4), …. need not be identical”.

Figure 4c-e Why do you not show all curves for all cell types?

We are a bit confused with this question. We do have the data corresponding to all the cell types mentioned in the figure.

On the other hand, if the reviewer means plotting the lines for all the cell types: we represented most data with symbols as plotting the lines for all the cell types makes the figure extremely hard to understand.

Figure 4d: Hexagons? Maybe stars?

Thanks for pointing this out. Actually these symbols are called Hexagrams that we incorrectly wrote Hexagons, we have corrected this in the revised manuscript.

p.10last paragraph of the main text is just a repetition of the previous paragraphs

We have now removed the repetitions. The last paragraph now summarises the achievements of the work.

Reviewer #2 (Recommendations for the authors):– The authors consider a regime where the area is equal to the target or preferred area of the cells and the area part of the free energy can be neglected. Can the validity of this approximation be made more quantitative? A naive guess would be that the modulus \λ_a is very large but in the end the authors calculate fluctuations of the area due to shape fluctuations and they would cost a large energy which is not taken into account. In the vertex model, as explained by the authors there is a transition between a soft solid and a hard solid behavior and the transition only depends on the ratio between the perimeter and the square root of the area. Which regime do the authors consider?

We thank the reviewer, Prof. Joanny, for the insightful comments, questions, and suggestions. We have included all the suggestions in the revised manuscript. Here, we briefly respond to the comments and questions.

There are two distinct analytical results in the paper: the distributions of aspect ratio and cell area. We have argued that the area constraint (or the constraint of confluency) is not crucial for the aspect ratio distribution, but it is extremely essential for the distribution of the area. As detailed below, we have made this argument more quantitative with additional results in the revise manuscript.

As we discussed in the main text, this approximation is motivated by our simulation results:

– First, as shown in Figure 2(c), the area distribution is sharply peaked around the average value. As the reviewer correctly points out, the sharpness of this distribution grows as *λ_A_* increases.

– Second, we find that even at moderate values of *λ_A_* (=0.1 — 0.8), where the area distribution is not a δ-function, the value of *α*, which determines the aspect ratio distribution, does not change (Figures 2b). We now show these distributions as well in the new supplementary Figure 2 —figure supplement 2.

– These results are consistent with the expectation that aspect ratio, being a dimensionless parameter, should be independent of the actual value of cellular area. Our simulation results, presented in Figure 2(b), and the additional results presented in the supplementary figures, Figure 2 —figure supplement 1 and Figure 2 —figure supplement 2, show that the area fluctuation does not significantly affect the shape fluctuation. Therefore, we are justified to neglect the area term in the energy, Equation (1), to obtain the aspect ratio distribution.

– On the other hand, as you also pointed out, for the other result, that is the area fluctuation due to shape fluctuation, there is indeed a large cost. We have taken this into account in an indirect manner (the discussion between Eqs. (20) and (21) in Sec. IV C), considering the very strong geometric constraint, that is the constraint of confluency. As you know, this is a hard mathematical problem and no exact results exist. We have used the results of Weaire et al. [Ref. 67] in the form of a constraint. This is a phenomenological approach, our analysis for the area distribution can be viewed as a support of this approach.

– We fixed the target perimeter, *P*_0_, in our simulations, except for the data in Figure 2(h), such that the system is in low-*P*_0_ regime. We believe that the low-*P*_0_ (hardsolid-like) regime is more relevant experimentally. But, as shown in Figure 2(h), the theory seems to capture the variation with *P*_0_ as well. We have now highlighted these points in the revised manuscript, and have specifically added a subsection (Two different regimes) in Appendix 2 discussing this.

– The authors define the tissue by an energy which is used in all the models that they quote. In my mind this energy is an efficient way to write that the forces in the epithelial layer are balanced. However, they then use a Boltzmann Boltzmann statistics with a well defined temperature for the configuration of the tissue. This temperature seems to play an important role as it enters the expression for the parameter \α. Is this equilibrium statistics also used in the simulations? In my mind, a tissue is an intrinsically non-equilibrium system and I do not see how the equilibrium distribution can be justified. The authors also claim that the results are not changed if cell division is considered, which is difficult for me to understand. The manuscript would be much stronger if the authors were discussing this effective equilibrium assumption in details.

The confluent models used in this work are well-established models for epithelial monolayers. These models have two distinct variants: depending on the presence or absence of activity. When there is no activity, the models are in equilibrium. The nonequilibrium models can have several interesting complex behaviours, as shown by you in the recent PRL paper (Ref. 73). But, the equilibrium models are also significant and capture the key static and dynamic properties of such systems, as shown by several groups including Jülicher, Fredberg, etc (Refs. 31, 52, 56, 65, etc.). How far can these equilibrium models represent a cellular system is an important question that we did not address in this work. This is also an important question in general. The role of activity is also significant, but for a cleaner understanding, we have not considered the effects of activity in the current paper. One crucial question we are currently exploring is whether an effective equilibrium behaviour is possible even under activity, similar to what we found in some other contexts of glassy dynamics. In fact, a single particle effective temperature description seems to capture many aspects in a dense glassy system [Nandi and Gov, Soft Matter (2017)]. We intend to explore this question in the future.

In the current work, in the absence of cell division and apoptosis, we are justified to use the Boltzmann distribution, as the temperature in these models has the same status as an equilibrium temperature. We did not consider any Boltzmann distribution in our simulations, but we now show in the additional data ( Appendix 1) that this is a reasonable approximation.

For the second point, we are sorry for the confusion. It is not yet clear to us how to develop an analytical description in the presence of cell-division and apoptosis, one possibility is to include these rates following this work: https://www.pnas.org/doi/ 10.1073/pnas.1011086107. We are also trying to apply the non equilibrium approach that you developed in the recent PRL (Ref. 73). Our point in the current manuscript is simply the following: when the rates of these processes are small, the basic characteristic of the system should remain similar. The analytical derivation suggests that the aspect ratio distribution will have two different parts: the algebraic part is determined by the geometric property that cell perimeter is closed-looped.

This cannot change. So, all system specific details should enter via *α* alone. What we showed in the manuscript is that this expectation seems to be reasonable when we include these processes in our simulations, with their rates being *not very large*. We have now clarified this in the manuscript (Page 9, Column 2, second paragraph): “We have neglected …. processes are not dominant (Appendix 4).”

– Another approximation is made in appendix A which replaces the square of the perimeter by the sum of the squares of the lengths of the edges of the cell. I understand that this strongly simplifies the calculations. Could the authors make more quantitative this approximation which is only justified by an inequality. How accurate is it?

Thanks for raising this question. This is indeed an important trick in our analytical derivation. We have now discussed this approximation in detail in Appendix 5 justifying this approximation. We have also discussed this point in Sec. IV A, Page 11: “Note that this inequality … integral slightly easier.”

– On more general grounds, the results obtained by the authors are interesting but do they have any biological relevance? Could the authors point out a biological phenomenon for which cell shape fluctuations would play an important role?

We believe these results can pave the way to understand pattern formation in a new light. Apart from providing explanations for several existing experimental results, the analytical result is important as we can now formulate a theoretical framework to understand the mechano-chemical pattern formation in confluent epithelial tissues in terms of this analytical result. Many past works have shown that cellular aspect ratio is directly correlated with biological functions, such as differentiation and division. But all these studies are either at the single cell level or in terms of the averages in a monolayer. Having the analytical distribution of the aspect ratio, we can now develop a statistical description of the monolayer. For example, we are now studying how to couple the model with different gradients of morphogens (such as the wingless and Dpp in the *Drosophila* wing disc). Having such a description will then allow us to study the pattern formation in such systems.

We have included a brief discussion of this aspect in the conclusion (Pages 9-10): “Pattern formation in a biological system … in the *Drosophila* wing disc.”